# Chapman-Enskog theory and crossover between diffusion and superdiffusion for nearly integrable quantum gases

**Maciej Łebek$^\star$ and Miłosz Panfil**

Faculty of Physics, University of Warsaw, Pasteura 5, 02-093 Warsaw, Poland

$\star$ maciej.lebek@fuw.edu.pl

## Abstract

Integrable systems feature an infinite number of conserved charges and on hydrodynamic scales are described by generalised hydrodynamics (GHD). This description breaks down when the integrability is weakly broken and sufficiently large space-time-scales are probed. The emergent hydrodynamics depends then on the charges conserved by the perturbation. We focus on nearly-integrable Galilean-invariant systems with conserved particle number, momentum and energy. Basing on the Boltzmann approach to integrability-breaking we describe dynamics of the system with GHD equation supplemented with a collision term. The limit of large space-time-scales is addressed using Chapman-Enskog expansion adapted to the GHD equation. For length scales larger than $\sim \lambda^{-2}$, where $\lambda$ is integrability-breaking parameter, we recover Navier-Stokes equations and find transport coefficients: viscosity and thermal conductivity. At even larger length scales, this description crosses over to Kardar-Parisi-Zhang universality class, characteristic to generic non-integrable one-dimensional fluids. Employing nonlinear fluctuating hydrodynamics we estimate this crossover length scale as $\sim \lambda^{-4}$.

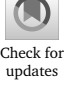

# 1   Introduction

Hydrodynamics is a universal theory of many-body systems out of equilibrium emerging at large space-time scales. It applies both to quantum and classical degrees of freedom and was proven extremely useful in systems as diverse as classical and quantum fluids [1–4], biological systems [5], relativistic matter [6], quantum circuits [7, 8] and electrically conducting fluids [9]. Within hydrodynamic approach, an enormous number of microscopic degrees of freedom is reduced to effective hydrodynamic fields, which follow the continuity equations written for densities of conserved quantities of the system.

From the perspective of the long history of hydrodynamics, the system which attracted the most attention was three-dimensional classical fluid (or gas), omnipresent in natural sciences. Basing on phenomenological arguments, it was established long time age that such system can be well described by the famous Navier-Stokes (NS) equations [2, 3]. NS equations take a universal form and non-universal details of the specific fluid are fixed by *transport coefficients*. In the context of NS equations, these are of two types: the first is fixed solely by the thermodynamics of the fluid, whereas the second is related to dissipative effects and requires knowledge about the microscopic dynamics of the particles. Dissipative coefficients of the three-dimensional fluid are shear and bulk viscosities and thermal conductivity. An extremely important challenge related to the NS theory is microscopic derivation of these quantities.

The microscopic explanation of NS equations was provided by the kinetic theory, with the Boltzmann equation as the central object of the framework. The first method which successfully connected collisional dynamics of Boltzmann equation to NS hydrodynamics was formulated by S. Chapman and D. Enskog over one hundred years ago. It has uncovered not only the universal NS equations by the means of suitable perturbative expansion but also pro-

vided quantative predictions for dissipative transport coefficients in terms of explicit integral equations, known today as Chapman-Enskog (ChE) integral equations.

Experimental [10–20] and theoretical progress [21–27] of the recent years gave rise to a large interest in dynamics of one-dimensional quantum integrable systems, such as spin chains or interacting bosonic and fermionic gases. Despite interactions between particles, integrable models possess an infinite number of conserved charges, displaying significantly different hydrodynamic behavior as compared to non-integrable counterparts. The relevant theory, which takes an infinite number of conservation laws into account, was developed in 2016 [28,29] and is known under the name of generalized hydrodynamics (GHD) [30–32]. It was tested in cold-atomic experiments [16,33] and nowadays GHD is the standard framework to tackle out-of-equilibrium dynamics of integrable systems.

Shortly after its formulation, the theory was extended in various directions. Notably, it was generalized to the case of external fields coupled to the charge densities [34,35] including perhaps the most important case of external potential coupled to density of particles. Moreover, the GHD theory was later extended to include the effects of diffusive broadening of quasiparticles due to interactions [36–38]. For the comprehensive review of recent advances in GHD theory, see the series of articles [39–46] with an introduction given in [47].

Integrable systems are fine-tuned models and are never exactly realized in experiments. Thus, it is important to study the effects of weak integrability-breaking [12, 39, 48–61], such as additional weak interactions between particles [62–65], which make the model non-integrable. In the context of large space-time-scale dynamics, these can be treated as corrections to the GHD dynamics of the underlying integrable model. It is established now that under effects of weak integrability-breaking the relevant equation for dynamics of quasiparticles is GHD equation supplemented with Boltzmann-like collision term [39,49,63,66], to which we will refer as GHD-Boltzmann equation. This feature renders the setup similar to the dynamics of classical Boltzmann equation, studied since a very long time.

The aim of the first part of this paper is to leverage this similarity and to adapt the ideas of Chapman and Enskog to a new physical context of Galilean-invariant nearly integrable quantum gases. In particular, as announced in a shorter paper [51], we derive the transport coeficients for such class of systems in terms of generalized ChE equations. Although GHD-Boltzmann equation differs from the standard Boltzmann counterpart by the presence of nonlinear effective velocity in the streaming term and by diffusion of quasiparticles, we show that a (generalized) ChE approach is capable of deriving the NS hydrodynamics from it, including the dissipative transport coefficients.

The results presented here parallel our developments reported in [51] where transport coefficients where derived from a linearized GHD-Boltzmann equation. Here, by generalizing the ChE method, we compute the transport coefficients beyond the linear approximation. Yet both approaches predict the same expressions for the transport coefficients, a feature recognized already in the classical kinetic theory [67] but a priori not guaranteed.

In the second part of our work, we address the important issue of the stability of NS equations due to interactions between hydrodynamic modes. Building on results of the first part, we employ the nonlinear fluctuating hydrodynamics (NLFH) to analyze the crossover between diffusive NS hydrodynamics and Kardar-Parisi-Zhang (KPZ) superdiffusion, which is expected to occur in generic non-integrable one-dimensional fluids [68–71] at largest length and time scales. Moreover, we estimate length scales on which different hydrodynamic descriptions (GHD, NS, KPZ) are valid.

The paper is organized as follows. Section 2 introduces the main ingredients of the framework that we use. These include Thermodynamic Bethe Ansatz (TBA), GHD-Boltzmann equation, collision integrals stemming from integrability-breaking, the role of Galilean invariance and length-scales relevant for the problem. In the following Section 3, we recall the ChE

theory for the classical Boltzmann equation. These first two sections serves as a basis for Section 4 in which we adapt the ChE method to the nearly integrable quantum gases described by the GHD-Boltzmann kinetic equation. In Section 5 we study equations of NLFH to analyze the crossover between NS diffusive and KPZ superdiffusive behavior of two-point functions. The paper finishes with conclusions presented in Sec. 6. Some additional formulas and short supplementary derivations are relegated to Appendices A and B.

## 2 Preliminaries

In this section we introduce the main objects used to derive the NS equations for nearly-integrable systems. We start with a short description of one-dimensional NS equations in 2.1. Next, we discuss the thermodynamics of quantum integrable models within the framework of the Thermodynamic Bethe Ansatz (TBA) and different representations of GHD-Boltzmann equation in 2.2. Consequences of Galilean invariance on the TBA quantities are derived in 2.3. After that in 2.4, we move to the discussion of the most important properties of collision integral in systems with conserved particle number, momentum and energy. In 2.5 we then derive mesoscopic conservation laws: the starting point for ChE expansion, which will be motivated by identification of small parameters in 2.6, which discussess the relevant lengthscales for our problem.

### 2.1 Navier-Stokes equations

The NS equations are continuity equations for conserved densities and associated currents. In $(1+1)$ dimensions these are particle density $\mathscr{q}_0 (= \varrho)$, momentum density $\mathscr{q}_1$ and energy density $\mathscr{q}_2$. Here and in the following, we suppress the dependence on the space-time coordinates $(x, t)$ unless its presence is required for the clarity. In the hydrodynamic treatment it is customary to use velocity $u$ and internal energy $e$ instead of $\mathscr{q}_1$ and $\mathscr{q}_2$. Furthermore, the internal energy density $e$ is often replaced by local temperature $T$ on the basis of the assumption of local thermal equilibrium. We will refer to the sets $\{\varrho, u, e\}$ or $\{\varrho, u, T\}$ as hydrodynamic fields.

The NS equations for the hydrodynamic fields, in the presence of the external potential $U(x)$ confining the particles, are [1–3]

$$\partial_t \varrho = -\partial_x(\varrho u), \quad \partial_t(\varrho u) = -\partial_x(\varrho u^2 + \mathcal{P}) - (\partial_x U)\varrho,$$
$$\partial_t(\varrho e) = -\partial_x(u \varrho e + \mathcal{J}) - \mathcal{P}\partial_x u. \tag{1}$$

The equations, beside the hydrodynamic fields, involve also hydrodynamic pressure $\mathcal{P}$ and heat current $\mathcal{J}$. In the phenomenological picture they take the following form

$$\mathcal{P} = P - \zeta \partial_x u, \qquad \mathcal{J} = -\kappa \partial_x T. \tag{2}$$

Here, we expressed the heat current $\mathcal{J}$ through a gradient of the temperature to conform with the standard formulation. The two transport coefficients are viscosity $\zeta$ and thermal conductivity $\kappa$, and the two relations in (2) are respectively the Newton's law for a viscous fluid (with shear viscosity absent in one spatial dimension) and the Fourier's law for the heat current. $P$ denotes here the standard thermodynamic pressure. The abovementioned thermodynamic relation between $e$ and $T$ fields which together with (2) allows to close the equations is [2]

$$dT = \frac{1}{c_V}de - \frac{1}{c_V}\left(\frac{P}{\varrho^2} - \frac{T}{\varrho^2}\left(\frac{\partial P}{\partial T}\right)_\varrho\right)d\varrho, \tag{3}$$

where $c_V$ is the specific heat at constant volume.

The NS equations (1) describe a Galilean invariant fluid. The density $\varrho$ and temperature $T$ are invariant (scalars) under Galilean boost, whereas velocity and internal energy transform in the standard way. Another invariants are pressure and transport coefficients. This implies that pressure $P \equiv P(\varrho, T)$ and transport coefficients depend only on the local density and temperature of the fluid $\zeta \equiv \zeta(\varrho, T)$, $\kappa \equiv \kappa(\varrho, T)$.

The central result of this work is derivation of the NS equations for nearly-integrable quantum gases described with a particle number, momentum and energy conserving Hamiltonian of the form

$$\hat{H} = \hat{H}_0 + \lambda \hat{V}, \qquad \lambda \ll 1, \tag{4}$$

where $\hat{H}_0$ is an interacting integrable Hamiltonian and $\hat{V}$ is the integrability-breaking term. Using a generalization of ChE method we recover exactly (1) with thermodynamic quantities such as specific heats and pressure dictated by the thermodynamics of the integrable model and with the explicit expressions for the two transport coefficients.

The transport coefficients that we find are sums of two contributions $\zeta = \zeta_{\mathcal{I}} + \zeta_{\mathfrak{D}}$ and $\kappa = \kappa_{\mathcal{I}} + \kappa_{\mathfrak{D}}$. The contributions $\zeta_{\mathfrak{D}}, \kappa_{\mathfrak{D}}$ originate from collisions of quasiparticles of the integrable model. They are an effect of interactions present in $H_0$ and are straightforwardly computed from the methods of integrability.[1] On the other hand, $\zeta_{\mathcal{I}}, \kappa_{\mathcal{I}}$ are related to Boltzmann collision integral $\mathcal{I}$ which represents the effects of integrability-breaking due to $\hat{V}$. Similarly as in the classical kinetic theory, computation of $\zeta_{\mathcal{I}}, \kappa_{\mathcal{I}}$ requires solution of integral equations. Expressions for transport coefficients were reported in [51] and we present their derivation in the ChE formalism in Sec. 4. In the classical kinetic theory there is only one type of contributions to the transport coefficients originating from the collision integral $\mathcal{I}$ since $H_0$ represents then a non-interacting system.

Finally, we mention an alternative representation of the NS equations in terms of the local conserved densities $\{\mathcal{q}_0, \mathcal{q}_1, \mathcal{q}_2\}$. These are related to the hydrodynamic fields through the following relations

$$\varrho = \mathcal{q}_0, \qquad u = \frac{\mathcal{q}_1}{\mathcal{q}_0}, \qquad e = \frac{\mathcal{q}_2}{\mathcal{q}_0} - \frac{\mathcal{q}_1^2}{2\mathcal{q}_0^2}. \tag{5}$$

The NS equations (1) for the conserved densities are then

$$\partial_t \mathcal{q}_0 + \partial_x \mathcal{q}_1 = 0,$$
$$\partial_t \mathcal{q}_1 + \partial_x \left( \frac{\mathcal{q}_1^2}{\mathcal{q}_0} + \mathcal{P} \right) = -(\partial_x U)\mathcal{q}_0,$$
$$\partial_t \mathcal{q}_2 + \partial_x \left( \frac{\mathcal{q}_1 \mathcal{q}_2}{\mathcal{q}_0} + \frac{\mathcal{q}_1}{\mathcal{q}_0}\mathcal{P} + \mathcal{J} \right) = -(\partial_x U)\mathcal{q}_1. \tag{6}$$

The two formulations of the NS equations will be useful in what follows.

## 2.2 Thermodynamic Bethe Ansatz and GHD-Boltzmann equation

In this section, we introduce the central object of our study – the GHD-Boltzmann equation. Together with it we introduce the main ingredients of the TBA. As we focus on Galilean invariant systems, the bare energy reads $E(\lambda) = \lambda^2/2$ and bare momentum is $p(\lambda) = \lambda$. We set the particle mass to unity $m = 1$.

The GHD equation, like a standard kinetic equation, is an equation for a (quasi-)particles' distribution $\rho_{\mathrm{p}}(\lambda; x, t)$. The integrable structure provides a number of alternative representations of the state of the system. The two additional useful representations are through the

---

[1]They are thermal matrix elements of the GHD diffusion operator $\mathfrak{D}$.

local values of conserved charges $\{q_n(x,t)\}$ and local (generalized) temperatures $\{\beta_n(x,t)\}$. The values of the former are given by

$$q_n(x,t) = \int d\lambda\, \hbar_n(\lambda)\rho_p(\lambda; x, t),\tag{7}$$

where $\hbar_n(\lambda)$ is the single-particle contribution to the charge density $q_n(x,t)$. For a Galilean-invariant theory the ultra-local charges are given by choosing

$$\hbar_n(\lambda) = \lambda^n/n!,\tag{8}$$

and charges are cumulants of the quasiparticles' distribution.

The second representation of the state is through the generalized temperatures and requires introducing the generalized TBA which we now quickly recall in the standard version applicable to homogeneous systems [72–75]. The initial object is the pseudoenergy $\epsilon_0$ which takes the following form

$$\epsilon_0(\lambda) = \sum_n \hbar_n(\lambda)\beta_n.\tag{9}$$

The local distribution of quasiparticles is then determined from the generalized TBA equations

$$\epsilon(\lambda) = \epsilon_0(\lambda) + \int d\mu\, \mathcal{T}(\lambda - \mu)F[\epsilon(\mu)],\tag{10}$$

$$\rho_{\text{tot}}(\lambda) = \frac{1}{2\pi} + \int d\mu\, \mathcal{T}(\lambda - \mu)n(\mu)\rho_{\text{tot}}(\mu),\tag{11}$$

where $\mathcal{T}(\lambda)$ is the model-dependent scattering kernel. For instance, for the Lieb-Liniger model [76,77] with coupling $c$ it reads $\mathcal{T}(\lambda) = \frac{c}{\pi}\frac{1}{c^2 + \lambda^2}$. Function $F[\epsilon(\mu)]$ is related to the free energy density $\mathsf{f} = \frac{1}{2\pi}\int d\lambda F[\epsilon(\lambda)]$ and depends on the statistics of particles [31]. For models with fermionic statistic of quasi-particles (as the Lieb-Linger model) it reads $F[\epsilon] = -\log(1 + e^{-\epsilon})$, for bosonic $F[\epsilon] = \log(1 - e^{-\epsilon})$ and for classical particles $F[\epsilon] = -e^{-\epsilon}$. Solving the first equation yields $\epsilon(\lambda)$ which enters the filling function

$$n(\lambda) = \left.\frac{dF}{d\epsilon}\right|_{\epsilon(\lambda)},\tag{12}$$

and in turn specifies the second equation. We note here that for the fermionic statistics $n(\lambda) = 1/(1 + e^{\epsilon(\lambda)})$. The total density $\rho_{\text{tot}}$ determines then the particles' density according to $\rho_p(\lambda) = n(\lambda)\rho_{\text{tot}}(\lambda)$.

Generalization to the non-homogeneous systems relies on assumption that locally the state of the system is described by the TBA with generalized temperatures varying in space. Namely the pseudoenergy (9) is promoted to

$$\epsilon_0(\lambda, x, t) = \sum_n \hbar_n(\lambda)\beta_n(x, t),\tag{13}$$

and the outcome of the TBA is space-time dependent $\rho_p(\lambda, x, t)$.

We move now to the GHD-Boltzmann equation for nearly integrable systems. It reads [35, 39, 50]

$$\partial_t \rho_p + \partial_x\left(v\rho_p\right) = (\partial_x U)\partial_\theta \rho_p + \frac{1}{2}\partial_x\left(\mathfrak{D}\partial_x \rho_p\right) + \mathcal{I}[\rho_p],\tag{14}$$

with effective velocity $v$, an external potential $U$, the diffusion kernel $\mathfrak{D}$ and the Boltzmann collision integral $\mathcal{I}[\rho_p]$. The effective velocity for Galilean-invariant models fulfills the following integral equation [28,31]

$$v(\lambda) = \hbar_1(\lambda) + 2\pi \int d\lambda'\, \mathcal{T}(\lambda - \lambda')\rho_p(\lambda')\left(v(\lambda') - v(\lambda)\right).\tag{15}$$

Before we describe the diffusion kernel $\mathfrak{D}$, we introduce some notation. For an arbitrary operator $\mathcal{O}$ we introduce the following notation for its action on arbitrary functions $\psi, \varphi$

$$(\mathcal{O}\psi)(\lambda) = \int d\mu \mathcal{O}(\lambda, \mu)\psi(\mu), \qquad \psi \mathcal{O} \varphi = \int d\lambda d\mu \mathcal{O}(\lambda, \mu)\psi(\lambda)\varphi(\mu), \qquad (16)$$

and use the standard notation for kernel multiplication

$$(\mathcal{O}_1 \mathcal{O}_2)(\lambda, \mu) = \int d\nu \mathcal{O}_1(\lambda, \nu)\mathcal{O}_2(\nu, \mu). \qquad (17)$$

The diffusion kernel is then (below $\rho_{\text{tot}}$ and $n$ should be understood as diagonal kernels) [37]

$$\mathfrak{D} = (1 - n\mathcal{T})^{-1}\rho_{\text{tot}}^{-1}\tilde{\mathfrak{D}}\rho_{\text{tot}}^{-1}(1 - n\mathcal{T}), \qquad \tilde{\mathfrak{D}}(\theta, \alpha) = \delta(\theta - \alpha)w(\theta) - W(\theta, \alpha), \qquad (18)$$

and

$$W(\theta, \alpha) = \rho_{\text{p}}(\theta)f(\theta)\left(\mathcal{T}^{\text{dr}}(\theta, \alpha)\right)^2 |v(\theta) - v(\alpha)|, \qquad w(\theta) = \int d\alpha W(\alpha, \theta), \qquad (19)$$

where $f(\theta)$ denotes statistical factor [31]

$$f(\theta) = -\frac{d^2 F/d\epsilon^2}{dF/d\epsilon}\bigg|_{\epsilon(\theta)}, \qquad (20)$$

which depends on the particles' statistics (it reads $f = 1 - n$ for fermions, $f = 1 + n$ for bosons and $f = 1$ for classical particles). The dressed scattering kernel fulfills

$$\mathcal{T}^{\text{dr}}(\lambda, \lambda') = \mathcal{T}(\lambda - \lambda') + \int d\lambda'' \mathcal{T}(\lambda - \lambda'')n(\lambda'')\mathcal{T}^{\text{dr}}(\lambda'', \lambda'), \qquad (21)$$

and in addition to that we define dressing of an arbitrary function $g(\lambda)$ as solution to the integral equation

$$g^{\text{dr}}(\lambda) = g(\lambda) + \int d\lambda' \mathcal{T}(\lambda - \lambda')n(\lambda')g^{\text{dr}}(\lambda'), \qquad (22)$$

which implies that $g^{\text{dr}} = (1 - \mathcal{T}n)^{-1}g$.

The effective velocity and the diffusion operator depend on the state $\rho_{\text{p}}$. This renders the GHD-Boltzmann equations non-linear even in the absence of the collision integral. The diffusion term leads to the entropy production [37] and in the interplay with the external potential leads to a stationary thermal state even in the absence of the collision term [78]. For now, we set $U = 0$ and discuss the effect of the external potential in Section 4.4.

In what follows, we will present a slightly different parametrization of the GHD-Boltzmann equation (14). We start with the relation between the generalized temperatures and conserved charges and associated currents, which are described by the hydrodynamic matrices

$$C_{ab} = -\frac{\partial q_a}{\partial \beta_b} = \hbar_a C \hbar_b, \qquad \mathcal{B}_{ab} = -\frac{\partial j_a}{\partial \beta_b} = \hbar_a \mathcal{B} \hbar_b, \qquad (23)$$

where $j_a$ is the expectation value of the conserved current operator $\hat{j}_a$ evaluated on the generalized TBA state [28],

$$j_a(x, t) = \int d\lambda \hbar_a(\lambda)v(\lambda; x, t)\rho_{\text{p}}(\lambda; x, t). \qquad (24)$$

The kernels of the two operators introduced above are

$$\mathcal{C} = (1 - n\mathcal{T})^{-1} \rho_{\mathrm{p}} f (1 - \mathcal{T}n)^{-1}, \qquad \mathcal{B} = (1 - n\mathcal{T})^{-1} \rho_{\mathrm{p}} f v (1 - \mathcal{T}n)^{-1}. \tag{25}$$

In the passing we note that hydrodynamic matrices (23) are symmetric. Moreover, the matrix $\mathcal{C}$ defines a *hydrodynamic inner product*, which we will use throughout the paper

$$(f|g) = f\mathcal{C}g. \tag{26}$$

After introducing the required ingredients we can now rewrite the GHD-Boltzmann equations. First, from the TBA equations it follows that [31, 51]

$$\frac{\partial \rho_{\mathrm{p}}}{\partial \beta_a} = -\mathcal{C}\hbar_a, \qquad \frac{\partial (v\rho_{\mathrm{p}})}{\partial \beta_a} = -\mathcal{B}\hbar_a. \tag{27}$$

These relations allow us to transform the GHD equations to equations for the generalized temperatures

$$\mathcal{C}_{ab}\partial_t\beta_b + \mathcal{B}_{ab}\partial_x\beta_b = \frac{1}{2}\partial_x\left((\mathfrak{D}\mathcal{C})_{ab}\partial_x\beta_b\right) - \hbar_a\mathcal{I}[\{\beta_b\}]. \tag{28}$$

We can also transform them into the continuity equations for the charges. The first two terms transform easily using again the hydrodynamic matrices,

$$\partial_t q_a = -\mathcal{C}_{ab}\partial_t\beta_b, \qquad \partial_x j_a = -\mathcal{B}_{ab}\partial_x\beta_b. \tag{29}$$

The diffusive term of the GHD equation requires expressing the change to $\beta_a$ due to a change to the conserved charges. Assuming the susceptibility matrix $\mathcal{C}_{ab}$ is invertible we have

$$\partial_x\beta_a = -\sum_b \mathcal{C}_{ab}^{-1}\partial_x q_b, \tag{30}$$

and

$$\partial_t q_a + \partial_x j_a = \frac{1}{2}\partial_x\left(\mathfrak{D}_{ab}\partial_x q_b\right) + \hbar_a\mathcal{I}[\{q_b\}]. \tag{31}$$

The three representations of the GHD-Boltzmann equation given in eqs. (14), (28) and (31) will be important for the discussion ahead.

## 2.3 Galilean invariance and thermal boosted state

We discuss now the role of the Galilean invariance. Specifically, we introduce the thermal boosted states and the transformations of the hydrodynamic matrices under the Galilean boost.

Consider a homogeneous state of the system described by distribution $\rho_{\mathrm{p}}$ which as a whole moves with velocity $u \neq 0$, where

$$u = \frac{\int \mathrm{d}\lambda \hbar_1(\lambda)\rho_{\mathrm{p}}(\lambda)}{\int \mathrm{d}\lambda \hbar_0(\lambda)\rho_{\mathrm{p}}(\lambda)}. \tag{32}$$

This state can be understood as a Galilean boost of the state with zero total momentum and with distribution function $\bar{\rho}_{\mathrm{p}}(\lambda)$ such that $\rho_{\mathrm{p}}(\lambda) = \bar{\rho}_{\mathrm{p}}(\lambda - u)$. If one distribution is a solution to the TBA equations so is the other one and the corresponding bare pseudo-energies obey the same type of relation $\epsilon_0(\lambda) = \bar{\epsilon}_0(\lambda - u)$. This relation is a consequence of the kernel $\mathcal{T}(\lambda - \mu)$ being a difference operator (which itself is required by the Galilean invariance). The transformation under the Galilean boost can be realized by introducing a boost operator $b_u$ such that $\rho_{\mathrm{p}} = b_u^{-1}\bar{\rho}_{\mathrm{p}}b_u$ with $b_0 = 1$ and $b_{-u} = b_u^{-1}$. In the rapidity variable the operators are represented by $b_u = \exp(u\partial_\lambda)$.

As discussed above, the TBA offers 3 equivalent characterizations of the GGE states one of which is the distribution function $\rho_{\mathrm{p}}$. Another one is provided by the generalized temperatures. The transformation laws for the generalized temperatures can be read off from the equality between the bare pseudo-energies $\epsilon_0(\lambda) = \bar{\epsilon}_0(\lambda - u)$. Expanding both sides of the equality in generalized temperatures we obtain

$$\sum_a \beta_a \hbar_a(\lambda) = \sum_a \bar{\beta}_a \hbar_a(\lambda - u). \tag{33}$$

Functions $\hbar_a(\lambda - u)$ expand in $\hbar_a(\lambda)$,

$$\hbar_a(\lambda - u) = \sum_b g_{ab}(-u)\hbar_b(\lambda), \tag{34}$$

where we introduced a transformation matrix $g_{ab}(-u)$ whose structure is a consequence of the monomial form of functions $\hbar_a(\lambda)$ defined in (8). From this it follows that

$$\beta_a = \sum_b \bar{\beta}_b g_{ba}(-u). \tag{35}$$

The transformation matrix $g(u)$ is a representation of the boost operator $b_u$ in the space of functions $\hbar_a$ determining the local conserved charges. The matrix is lower triangular because $\hbar_a(\lambda - u)$ expands into $\hbar_b(\lambda)$ with $b \leq a$. We also have

$$g(0) = 1, \qquad g^{-1}(u) = g(-u). \tag{36}$$

In principle, $g(u)$ is an infinite-dimensional matrix, but because of its triangular structure it can be truncated at a chosen order. For example, in the subspace of 3 lowest conserved charges, which will be relevant in deriving the NS equations, the matrix is

$$g(u) = \begin{pmatrix} 1 & 0 & 0 \\ u & 1 & 0 \\ \frac{1}{2}u^2 & u & 1 \end{pmatrix}, \tag{37}$$

and its action in this subspace is closed and is invertible.

The expectation values of the conserved charges transform in the similar way as $\hbar_a$,

$$\mathscr{q}_a = \sum_b g_{ab}(u)\bar{\mathscr{q}}_b, \tag{38}$$

where $\bar{\mathscr{q}}_a$ and $\mathscr{q}_a$ are values before and after the boost respectively. We also note that the effective velocity transforms as

$$v(\lambda) = \bar{v}(\lambda - u) + u. \tag{39}$$

For the analysis of the transport coefficients the transformation properties of the hydrodynamic matrices are important. First we note that, at the level of operators, we have

$$\mathcal{C}(\mu, \lambda) = b_u^{-1}\bar{\mathcal{C}}(\mu, \lambda)b_u, \qquad \mathcal{B}(\mu, \lambda) = b_u^{-1}\bar{\mathcal{B}}(\mu, \lambda)b_u + u b_u^{-1}\bar{\mathcal{C}}(\mu, \lambda)b_u, \tag{40}$$

with the operators acting now on both variables, $b_u \sim e^{u\partial_\mu} \otimes e^{u\partial_\lambda}$. This implies that

$$\begin{aligned}
\mathcal{C}_{ab} &= \sum_{cd} g_{ac}(u)\bar{\mathcal{C}}_{cd}(g^T(u))_{db} = \left(g(u)\bar{\mathcal{C}}g^T(u)\right)_{ab}, \\
\mathcal{B}_{ab} &= \left(g(u)\bar{\mathcal{B}}g^T(u)\right)_{ab} + u\left(g(u)\bar{\mathcal{C}}g^T(u)\right)_{ab}.
\end{aligned} \tag{41}$$

Furthermore and independently of the boost, $\mathcal{C}_{a0} = \mathcal{B}_{a1}$. This follows from the fact [31] that $v(\lambda) = \hbar_1^{\mathrm{dr}}(\lambda)/\hbar_0^{\mathrm{dr}}(\lambda)$ and from definitions (25) of $\mathcal{C}, \mathcal{B}$ kernels.

Among all the states described by the generalized TBA a special role is played by thermal states and boosted thermal states. The former are described by non-zero $\bar{\beta}_0$ and $\bar{\beta}_2$ with all other chemical potentials zero. Boosting the thermal state by $u$ the generalized temperatures become

$$\beta_0 = \bar{\beta}_0 + \frac{1}{2}u^2\bar{\beta}_2\,, \qquad \beta_1 = -u\bar{\beta}_2\,, \qquad \beta_2 = \bar{\beta}_2\,. \tag{42}$$

This implies that (inverse) temperature is an invariant of the thermal state under the Galilean boost. Another invariant is the particle's density $\mathcal{q}_0$. Therefore, a triple $(\mathcal{q}_0, \mathcal{q}_1, \beta_2)$ provides a convenient parametrization of thermal boosted states. It transforms minimally under the Galilean boost by $u$ to $(\mathcal{q}_0, \mathcal{q}_1 + u, \beta_2)$.

The particles distribution for a thermal state is an even function, $\rho_{\mathrm{p}}(\lambda) = \rho_{\mathrm{p}}(-\lambda)$. A consequence of this is a checker-board structure of the hydrodynamic matrices. In the subspace of the 3 first conserved charges

$$\bar{\mathcal{C}}_{ab} = \begin{pmatrix} \bar{\mathcal{C}}_{00} & 0 & \bar{\mathcal{C}}_{02} \\ 0 & \bar{\mathcal{C}}_{11} & 0 \\ \bar{\mathcal{C}}_{02} & 0 & \bar{\mathcal{C}}_{22} \end{pmatrix}\,, \qquad \bar{\mathcal{B}}_{ab} = \begin{pmatrix} 0 & \bar{\mathcal{B}}_{01} & 0 \\ \bar{\mathcal{B}}_{01} & 0 & \bar{\mathcal{B}}_{12} \\ 0 & \bar{\mathcal{B}}_{12} & 0 \end{pmatrix}\,. \tag{43}$$

We observe that $\bar{\mathcal{C}}_{ab}$ is invertible as $\det\bar{\mathcal{C}} = \bar{\mathcal{C}}_{11}(\bar{\mathcal{C}}_{00}\bar{\mathcal{C}}_{22} - \bar{\mathcal{C}}_{02}) = T^4\varrho^4 c_V \varkappa_T$, where $\varkappa_T$ is isothermal compressibility of the integrable model and $c_V$ is specific heat at constant volume. Thermodynamic stability conditions guarantee that $c_V, \varkappa_T > 0$, hence $\det\bar{\mathcal{C}} > 0$. Above we have used formulas for thermodynamic quantities expressed with matrix elements of $\bar{\mathcal{C}}, \bar{\mathcal{B}}$, given in Appendix A.1. The hydrodynamic matrices for thermal boosted states follow then from transformations (41).

## 2.4 Collision integral

In this section we look more closely at the collision integral $\mathcal{I}[\rho_{\mathrm{p}}]$ and discuss the most important properties of it. The collision integrals considered by us originate from integrability-breaking. In Ref. [35,63] it was shown how such terms arise due to perturbative treatment at the level of Fermi's Golden Rule (FGR). However, the ChE method does not depend on the precise form of the collision integral. Instead, it relies on two general and physically motivated properties that we now discuss. We refer to [51] for an example of a collision integral in the system of two coupled Lieb-Liniger models. It can be explicitly checked that collision integrals derived in [35,63] have the properties listed below.

First of all, a characteristic feature of the collision integral $\mathcal{I}[\rho_{\mathrm{p}}]$ is a set of collision invariants, namely quantities conserved by it. The natural choice for the collision invariants are the particle number, momentum and energy and such choice yields the NS equations. For collision integral $\mathcal{I}$ respecting these three conservation laws we have

$$\hbar_a \mathcal{I} = \int \mathrm{d}\theta\, \hbar_a(\theta)\mathcal{I}[\rho_{\mathrm{p}}](\theta) = 0\,, \qquad a = 0,1,2\,. \tag{44}$$

Another choice, for example, is to respect only the particle number, the resulting hydrodynamic equations are the non-linear diffusion equations. We will investigate such situation in a future work.

The second important feature of the collision integral are the stationary states $\rho_{\mathrm{p}}^{\mathrm{st}}$ fulfilling

$$\mathcal{I}[\rho_{\mathrm{p}}^{\mathrm{st}}] = 0\,. \tag{45}$$

We will generally assume that the stationary states are TBA boosted thermal states described in the section above. This can be shown for classical Boltzmann collision kernel [2]. Moreover, in Ref. [35] it was demonstrated for collision integrals constructed from FGR that indeed the stationary states are TBA boosted thermal states with pseudoenergies of the form

$$\epsilon_0(\theta) = -\beta\mu + \nu\hbar_1(\theta) + \beta\hbar_2(\theta). \tag{46}$$

However, collision integrals might also posses non-thermal stationary states as discussed, for example, in [65]. Such situations are also beyond the scope of the present work.

In our calculations we will be mostly interested in the operator

$$\Gamma = -\frac{\delta\mathcal{I}}{\delta\rho_{\mathrm{p}}}\mathcal{C}. \tag{47}$$

The conservation of energy, momentum and particle number by the collision integral implies that $\Gamma$ has three zero eigenvectors

$$\int \mathrm{d}\lambda\, \hbar_a(\lambda)\Gamma(\lambda,\mu) = 0, \qquad a = 0, 1, 2. \tag{48}$$

In our work, we also assume that $\Gamma$ beside these three zero eigenvalues, the rest of the spectrum is positive and gapped, with a gap $\gamma > 0$ providing a characteristic time-scale for thermalization in the system. These assumptions were confirmed in a concrete example of coupled Lieb-Liniger models investigated in [51].

## 2.5 Mesoscopic conservation laws

In this section we make the first steps towards recovering hydrodynamic equations from GHD-Boltzmann equation (14). We start by writing the equations for the dynamics of the first three charges. Existence of three collision invariants leads to three mesoscopic conservation laws, c.f. (31),

$$\partial_t q_a + \partial_x j_a = \frac{1}{2}\partial_x(\mathfrak{D}_{ab}\partial_x q_b), \qquad a = 0, 1, 2, \tag{49}$$

with $j_a(x,t)$ defined in (24).

This set of three equations is not closed. The equations depend on the full information of the state of the system, not only on the first three charge densities. Otherwise, these equations are similar to the NS equations in a proper representation. The ChE method allows us to effectively integrate out the information about the evolution of the other charges which are not conserved by the collision term. Because of that, they are decaying and lead to appearance of additional (beside the contribution from the GHD diffusion) dissipative terms.

It is helpful to write equations (49) more explicitly. First, we use the Markovianity property of the diffusion operator. It implies that $\hbar_0\mathfrak{D} = 0$ for Galilean-invariant systems [36] and the density obeys a conservation laws without diffusive terms,

$$\partial_t q_0 + \partial_x j_0 = 0, \qquad j_0 = q_1. \tag{50}$$

The other two continuity equations are

$$\begin{aligned}\partial_t q_1 + \partial_x j_1 &= -\partial_x\mathcal{P}_{\mathfrak{D}},\\ \partial_t q_2 + \partial_x j_2 &= -\partial_x(u\mathcal{P}_{\mathfrak{D}} + \mathcal{J}_{\mathfrak{D}}),\end{aligned} \tag{51}$$

where we defined

$$\mathcal{P}_{\mathfrak{D}} = -\frac{1}{2}\int \mathrm{d}\lambda\mathrm{d}\mu\, \hbar_1(\lambda)\mathfrak{D}(\lambda,\mu)\partial_x\rho_{\mathrm{p}}(\mu) = -\hbar_1\mathfrak{D}\partial_x\rho_{\mathrm{p}}/2, \tag{52}$$

$$u\mathcal{P}_{\mathfrak{D}} + \mathcal{J}_{\mathfrak{D}} = -\frac{1}{2}\int \mathrm{d}\lambda\mathrm{d}\mu\, \hbar_2(\lambda)\mathfrak{D}(\lambda,\mu)\partial_x\rho_{\mathrm{p}}(\mu) = -\hbar_2\mathfrak{D}\partial_x\rho_{\mathrm{p}}/2, \tag{53}$$

anticipating the structure of the expressions.

The currents $j_a$ involve two types of contributions. The first one depends only on the hydrodynamic fields, the second requires knowledge of the full rapidity distribution $\rho_p$ via contributions to hydrodynamic pressure $\mathcal{P}_\nu$ and heat current $\mathcal{J}_\nu$. The result is

$$j_1 = \frac{q_1^2}{q_0} + \mathcal{P}_\nu, \qquad j_2 = \frac{q_1 q_2}{q_0} + u\mathcal{P}_\nu + \mathcal{J}_\nu, \tag{54}$$

where we defined the contributions to pressure and heat current

$$\mathcal{P}_\nu = \int d\lambda\,(\lambda - u)v(\lambda)\rho_p(\lambda), \tag{55}$$

$$\mathcal{J}_\nu = \frac{1}{2}\int d\xi\,\xi^2(v(u+\xi)-u)\rho_p(u+\xi) = \frac{1}{2}\int d\lambda(\lambda-u)^2(v(\lambda)-u)\rho_p(\lambda). \tag{56}$$

The resulting equations are now

$$\partial_t q_0 + \partial_x q_1 = 0,$$
$$\partial_t q_1 + \partial_x\left(\frac{q_1^2}{q_0} + \mathcal{P}\right) = 0, \tag{57}$$
$$\partial_t q_2 + \partial_x\left(\frac{q_1 q_2}{q_0} + u\mathcal{P} + \mathcal{J}\right) = 0.$$

They take a form of the NS equations (1) with $\mathcal{P} = \mathcal{P}_\nu + \mathcal{P}_\mathfrak{D}$ and $\mathcal{J} = \mathcal{J}_\nu + \mathcal{J}_\mathfrak{D}$. Anticipating our findings, the ChE method approximates expressions for $\mathcal{P}$ and $\mathcal{J}$ such that they depend only on the hydrodynamic fields through Newton's and Fourier's laws (2), thus closing the set of equations.

## 2.6 Relevant lengthscales

In the GHD-Boltzmann equation we have (at least) 3 relevant scales in the system. There is a microscopic length-scale $l_\mathfrak{D}$ (given by the scattering shift of the integrable theory), there is a hydrodynamic length-scale $l_h$ at which $\rho_p$ varies in space and there is time-scale $\tau_\mathcal{I}$ specyfing the rate of the integrability-breaking collisions. Introducing the latter into the GHD equations we obtain (below we defined $\mathcal{I} = \frac{1}{\tau_\mathcal{I}}\tilde{\mathcal{I}}$)

$$\partial_t\rho_p + \partial_x(v\rho_p) - \frac{1}{2}\partial_x(\mathfrak{D}\partial_x\rho_p) = \frac{1}{\tau_\mathcal{I}}\tilde{\mathcal{I}}[\rho_p], \tag{58}$$

with the following dimensions

$$[\rho_p] = [\tilde{\mathcal{I}}] = \frac{1}{m^2/s}, \qquad [\tau_\mathcal{I}] = s, \qquad [v] = m/s, \qquad [\mathfrak{D}] = m^2/s. \tag{59}$$

We introduce also $\langle v\rangle$, a typical velocity of quasiparticles which is a characterstic of the state. Then $l_\mathcal{I} = \langle v\rangle\tau_\mathcal{I}$ is a distance travelled by a quasiparticle between the non-integrable collisions. We can now rewrite the GHD equations in a dimensionless form. To this end, we write

$$\tilde{v} = \frac{v}{\langle v\rangle}, \qquad \tilde{\mathfrak{D}} = \frac{\mathfrak{D}}{\langle v\rangle l_\mathfrak{D}}, \tag{60}$$

which defines the dimensionless effective velocity and diffusion operator. Then, in the reduced variables $\tilde{t} = t/\tau_h$ and $\tilde{x} = x/l_h$, we find

$$\partial_{\tilde{t}}\rho_p + \partial_{\tilde{x}}(\tilde{v}\rho_p) - \frac{1}{2}\frac{\tau_\mathfrak{D}}{\tau_h}\partial_{\tilde{x}}(\bar{\mathfrak{D}}\partial_{\tilde{x}}\rho_p) = \frac{\tau_h}{\tau_\mathcal{I}}\tilde{\mathcal{I}}[\rho_p]. \tag{61}$$

The two ratios appearing in the equation can be reinterpreted through length-scales. Introducing parameters which can be identified as generalizations of Knudsen numbers [2] from standard kinetic theory

$$\delta_{\mathfrak{D}} = \frac{\tau_{\mathfrak{D}}}{\tau_{\mathrm{h}}} = \frac{l_{\mathfrak{D}}}{l_{\mathrm{h}}}, \qquad \delta_{\mathcal{I}} = \frac{\tau_{\mathcal{I}}}{\tau_{\mathrm{h}}} = \frac{l_{\mathcal{I}}}{l_{\mathrm{h}}}, \tag{62}$$

the GHD-Boltzmann equation becomes

$$\partial_{\tilde{t}}\rho_{\mathrm{p}} + \partial_{\tilde{x}}(\tilde{v}\rho_{\mathrm{p}}) - \frac{1}{2}\delta_{\mathfrak{D}}\partial_{\tilde{x}}(\tilde{\mathfrak{D}}\partial_{\tilde{x}}\rho_{\mathrm{p}}) = \frac{1}{\delta_{\mathcal{I}}}\tilde{\mathcal{I}}[\rho_{\mathrm{p}}]. \tag{63}$$

We note that in order to be in the hydrodynamic regime we require $\delta_{\mathfrak{D}} \ll 1$ and in order to treat the collision integral as a perturbation we need also $\delta_{\mathcal{I}} \ll 1$. The second condition guarantees that collisions occur in effectively homogeneous background and system thermalizes locally.

Lastly, it is useful to realize that almost identical set of mesoscopic conservation laws as (57) can be derived in the rescaled units. What happens is that we recover the same equations with a single difference that parameter $\delta_{\mathfrak{D}}$ appears explicitly in expressions for $\mathcal{P}_{\mathfrak{D}}$ and $\mathcal{J}_{\mathfrak{D}}$

$$\mathcal{P}_{\mathfrak{D}} = -\delta_{\mathfrak{D}}\hbar_1 \mathfrak{D}\partial_x \rho_{\mathrm{p}}/2, \qquad u\mathcal{P}_{\mathfrak{D}} + \mathcal{J}_{\mathfrak{D}} = -\delta_{\mathfrak{D}}\hbar_2 \mathfrak{D}\partial_x \rho_{\mathrm{p}}/2. \tag{64}$$

In what follows we work with the rescaled units (but without using the *tilde* symbol explicitly) which introduces small parameters $\delta_{\mathfrak{D}}, \delta_{\mathcal{I}}$. After truncating expansion at a given order, as a last step we will restore the original units, thus eliminating the generalized Knudsen numbers from the equations.

## 3 Chapman-Enskog theory for the Boltzmann equation

ChE expansion [2–4] was the first theory, which successfully connected microscopic dynamics of Boltzmann equation to the equations of hydrodynamics. It was developed by S. Chapman and D. Enskog more than one hundred years ago. Originally formulated for three-dimensional classical gas, it has uncovered not only the universal NS equations but also provided quantative predictions for transport coefficients in terms of explicit integral equations. In this section we write a summary of this classic result. It provides context for the generalization of the method to quantum integrable theories developed in the next section.

We start with the Boltzmann equation written as

$$\partial_t \rho + v\partial_x \rho = I_B[\rho], \tag{65}$$

where $\rho(x, v)$ is one-particle distribution function on a phase space and $I_B[\rho]$ is the Boltzmann collision integral, a nonlinear functional of the $\rho$ distribution. For the sake of this discussion, we do not discuss the of the operator $I_B$ and consider one-dimensional system. The main idea behind ChE expansion is based on two important observations about the Boltzmann equation:

- The only stationary states of collision integral $I_B[\rho]$ are space-varying boosted thermal states. For classical gas these are given by boosted Maxwell-Boltzmann distributions

$$\rho_{\mathrm{bth}}(x, v) = \frac{\varrho(x)}{\sqrt{2\pi T(x)}} \exp\left(-\frac{(v - u(x))^2}{2T(x)}\right), \tag{66}$$

  where $\varrho(x), u(x), T(x)$ are density, velocity and temperature of the gas at the point $x$, respectively. We have set $m = 1$ and $k_B = 1$. The process of relaxation from a generic state towards local boosted thermal states happens on a characteristic *relaxation*

*time-scale* $\tau_{I_B}$, which depends on the interaction potential between particles encoded in collision integral $I_B[\rho]$ and on the state itself.

We emphasize here that in general the states (66) are stationary states of collision integral term only and some dynamics is generated by the streaming term $v\partial_x\rho$. Global equilibrium states of (65) correspond to boosted thermal states with hydrodynamic fields uniform in space $\varrho(x), u(x), T(x) = \text{const}$, for which the streaming term vanishes as well.

- Collision integral conserves density of particles, momentum and energy. This is equivalent to the existence of three collision invariants $\hbar_0(v) = 1$, $\hbar_1(v) = v$, $\hbar_2(v) = \frac{1}{2}v^2$, such that

$$\int dv\, \hbar_a(v)I_B[\rho](v) = 0\,, \qquad a = 0, 1, 2\,. \tag{67}$$

Naturally, collision invariants can be directly related to hydrodynamic fields $\varrho(x)$, $u(x)$, $e(x)$, which are given by

$$\varrho(x) = \langle \hbar_0 \rangle\,, \qquad u(x) = \frac{\langle \hbar_1 \rangle}{\langle \hbar_0 \rangle}\,, \qquad e(x) = \frac{\langle \hbar_2 \rangle}{\langle \hbar_0 \rangle} - \frac{1}{2}u(x)^2\,, \tag{68}$$

where $\langle \cdot \rangle$ denotes average over velocities with respect to $\rho$. What is more, the temperature field is simply given by $e(x) = \frac{1}{2}T(x)$, assuming that the system is in local thermodynamic equilibrium. We collectively denote the hydrodynamic fields as $\{\varrho_a(x, t)\}$.

The observations listed above point towards the existence of two characteristic timescales in the system. Starting from an initial distribution, on a timescale given by $\tau_{I_B}$ the system will *locally thermalize*, such that it can be described by hydrodynamic fields $\varrho(x), u(x)$ and $T(x)$. On a longer, hydrodynamic timescale $\tau_h$ the streaming term will be dominant and these fields will evolve according to the standard equations of hydrodynamics. In order to see their structure explicitly, we multiply the Boltzmann equation with collision invariants and integrate over $dv$, similarly as in Sec. 2.5. We recover three equations which read

$$\partial_t \varrho + \partial_x(\varrho u) = 0\,,$$
$$\partial_t(\varrho u) + \partial_x\left(\varrho u^2 + \mathcal{P}\right) = 0\,, \tag{69}$$
$$\partial_t(\varrho e) + \partial_x(u\varrho e + \mathcal{J}) + \mathcal{P}\partial_x u = 0\,,$$

with dynamical pressure and heat current

$$\mathcal{P} = \int dv\,(v - u)v\rho(v)\,, \tag{70}$$

$$\mathcal{J} = \frac{1}{2}\int dv(v - u)^3\rho(v)\,. \tag{71}$$

The hydrodynamic equations presented above are not closed as the dynamical pressure and heat current depend on the whole distribution $\rho$, not only on its first three moments (or equivalently, on hydrodynamic fields). The closure of equation will be the central point of the ChE expansion procedure, which we now discuss.

The two observations about Boltzmann equation listed before can be turned into perturbative expansion. To formally motivate it, we start with estimating the order of terms in (65). We assume that the state varies at a hydrodynamic length-scale $l_h$ and the particles have some characteristic velocity $\bar{v}$. Then the hydrodynamic timescale associated to the streaming term

is $\tau_{\mathrm{h}} = l_{\mathrm{h}}/\bar{v}$. In the dimensionless units $\tilde{x} = x/l_{\mathrm{h}}$, $\tilde{t} = t/\tau_{\mathrm{h}}$, $\tilde{v} = v/\bar{v}$ and explicitly extracting the timescale from collision term $\tilde{I}_B = \frac{1}{\tau_{I_B}} I_B$ we can write Boltzmann equation as

$$\partial_{\tilde{t}} \rho + \tilde{v} \partial_{\tilde{x}} \rho = \frac{1}{\delta} \tilde{I}_B[\rho],\tag{72}$$

where $\delta = \tau_{I_B}/\tau_{\mathrm{h}}$ defines the so-called Knudsen number [2]. We wish to address hydrodynamic limit of slow (in comparison to microscopic scales) space and time variation, hence Knudsen number is considered to be small $\delta \ll 1$. The form of Boltzmann equation (72) suggests perturbative solution in $\delta$ and indeed, in the ChE method we essentially assume that

$$\rho(x,v;t) = \rho^{(0)}(x,v;t)\left(1 - \sum_{n=1}^{\infty} \delta^n \rho^{(n)}(x,v;t)\right).\tag{73}$$

However, the expansion cannot be considered to be fully systematic as it is supplemented by additional assumptions which we will discuss now. Before that and in what follows we drop tildes from the notation.

Firstly, the crucial aspect of the method is the assumption about the time evolution of distribution $\rho$. We assume that it depends on time through three time-dependent hydrodynamic fields collectively denoted $\{\varrho_a(x,t)\}$,

$$\rho(x,v;t) \to \rho(x,v|\{\varrho_a(x,t)\}).\tag{74}$$

Moreover, there is an assumption about expansion which can be formulated as the following conditions

$$\langle \hbar_a \rangle = \int \mathrm{d}v\, \hbar_a(v) \rho^{(0)}(x,v), \qquad 0 = \int \mathrm{d}v \rho^{(0)}(x,v)\, \hbar_a(v) \rho^{(n)}(x,v) = (\hbar_a|\rho^{(n)}),\tag{75}$$

for $n \geq 1$, which means that the averages of collision invariants (and in consequence, hydrodynamic fields) are given solely by the zeroth order contribution to the distribution $\rho^{(0)}$. In other words, we essentially assume that the hydrodynamic fields are not expanded in $\delta$ and corrections $\{\rho^{(n)}\}_{n\geq 1}$ are orthogonal to collision invariants (in the sense of the hydrodynamic inner product $(\cdot|\cdot)$, see (26), note that for classical particles $\mathcal{C} = \rho$). However, we crucially assume that dynamical pressure and heat current (70) do admit expansion in $\delta$

$$\mathcal{P} = \sum_{n=0}^{\infty} \delta^n \mathcal{P}^{(n)}, \qquad \mathcal{J} = \sum_{n=0}^{\infty} \delta^n \mathcal{J}^{(n)}.\tag{76}$$

Expanding now the problem to different orders in $\delta$ yields hydrodynamic equations at different scales.

## 3.1 Euler scale

We proceed now to (72) and plug the expansion (73). At zero-th order in $\delta$ we find

$$I_B[\rho^{(0)}] = 0.\tag{77}$$

From the properties of the scattering integral this equation implies that the distribution is locally thermal

$$\rho^{(0)}(x,v;t) = \rho_{\mathrm{bth}}(x,v|\{\varrho_a(t)\}).\tag{78}$$

With this solution, we may evaluate

$$\mathcal{P}^{(0)} = P = \varrho T, \qquad \mathcal{J}^{(0)} = 0,\tag{79}$$

where by $P$ we have denoted the hydrostatic pressure of ideal gas. Moreover, at this order of expansion the hydrodynamic equations take the form of *Euler equations*

$$
\begin{aligned}
\partial_t \varrho &= -\partial_x(\varrho u)\,, \\
\partial_t u &= -u\partial_x u - \frac{T}{\varrho}\partial_x \varrho - \partial_x T\,, \\
\partial_t T &= -2T\partial_x u - u\partial_x T\,,
\end{aligned}
\tag{80}
$$

which do not lead to entropy production in the system [3]. We point out that these are Euler equations with thermodynamics of ideal gas, even though interactions encoded in Boltzmann term were necessary for their emergence.

### 3.2 Navier-Stokes scale

We go now to the first order in $\delta$. We find an equation

$$
\partial_t \rho^{(0)}(x,v;t) + v\partial_x \rho^{(0)}(x,v;t) = [\Gamma_B \rho^{(1)}](x,v;t)\,,
\tag{81}
$$

where $\Gamma_B = -(\delta I_B/\delta\rho)|_{\rho^{(0)}}\,\rho^{(0)}$. More explicitly we have

$$
\sum_a \frac{\partial \rho^{(0)}}{\partial \varrho_a} G_a^{(0)}(\{\varrho_b\}) + v\sum_a \frac{\partial \rho^{(0)}}{\partial \varrho_a}\partial_x \varrho_a = [\Gamma_B \rho^{(1)}](x,v;t)\,,
\tag{82}
$$

where derivatives $\frac{\partial \rho^{(0)}}{\partial \varrho_a}$ can be explicitly computed. Moreover, the operator $G_a^{(n)}(\{\varrho_b\})$ defines the dynamics of hydrodynamic field $\varrho_a$ at $n$-th order in $\delta$ (dependence on $\delta$ enters via hydrodynamic pressure and heat current)

$$
\partial_t \varrho_a = \sum_{n=0}^{\infty} \delta^n G_a^{(n)}(\{\varrho_b\})\,.
\tag{83}
$$

The expressions for $G_a^{(0)}(\{\varrho_b\})$ can be simply read off from (80). After short calculation we get

$$
\rho^{(0)}(x,v)\left(\frac{u(x)-v}{T(x)^2}\left(\frac{3}{2}T(x) - \frac{(v-u(x))^2}{2}\right)\partial_x T\right) = [\Gamma_B \rho^{(1)}](x,v)\,.
\tag{84}
$$

We evaluate this equation at $v = \xi + u$. In the relative $\xi$ variable, the equation can be expressed with local unboosted thermal state $\bar\rho^{(0)}(\xi) = \rho^{(0)}(\xi + u)$ (corresponding to the same hydrodynamic fields $\varrho(x), T(x)$ but with $u(x) = 0$) and $\bar\rho^{(1)}(\xi) = \rho^{(1)}(\xi + u)$. We then find

$$
\bar\rho^{(0)}(\xi)\frac{\xi}{T^2}\left(\frac{\xi^2}{2} - \frac{3}{2}T\right)\partial_x T = [\bar\Gamma_B \bar\rho^{(1)}](\xi)\,.
\tag{85}
$$

Transformation between Eqs. (84) and (85) is in fact a Galilean boost and illustrates Galilean invariance of the theory and as a result of the transport coefficients.

Assuming that $\bar\rho^{(1)}(\xi) = \bar\phi(\xi)\partial_x T$, we find that function $\bar\phi(\xi)$ fulfills an integral equation

$$
[\bar\Gamma_B \bar\phi](\xi) = \chi(\xi)\,, \qquad \chi(\xi) = \bar\rho^{(0)}(\xi)\frac{\xi}{T^2}\left(\frac{\xi^2}{2} - \frac{3}{2}T\right)\,,
\tag{86}
$$

with additional conditions (75)

$$
(\hbar_a|\bar\phi) = 0\,, \qquad a = 0,1,2\,.
\tag{87}
$$

We note that the function $\bar{\phi}(\xi)$ is antisymmetric (this follows from antisymmetry of $\chi(\xi)$ and properties of $\bar{\Gamma}_B$) and compute corrections to hydrodynamic pressure and heat current finding

$$\mathcal{P}^{(1)} = 0, \qquad \mathcal{J}^{(1)} = -\frac{1}{2}\int d\xi \bar{\rho}^{(0)}(\xi)\xi^3 \bar{\phi}(\xi)\partial_x T, \tag{88}$$

recovering Newton's and Fourier's laws (2) with

$$\eta = 0, \qquad \kappa = \frac{1}{2}\int d\xi \bar{\rho}^{(0)}(\xi)\xi^3 \bar{\phi}(\xi). \tag{89}$$

We have thus found NS equations with vanishing bulk viscosity. This is a known result for a weakly perturbed ideal gas [2,3]. In order to evaluate thermal conductivity $\kappa$ one has to find $\bar{\phi}(\xi)$ which involves solving the integral equation (86).

The expansion in $\delta$ can be carried out to higher orders which leads to so-called Burnett hydrodynamic equations [2]. We close our discussion at the first order and point out that the found expressions for transport coefficients (89) can be understood as special cases of the general formulas (143) (or (A.18)) derived by us in this work. One can recover them by plugging non-interacting limits of hydrodynamic matrices, diffusion operator and thermodynamic quantities computed for ideal gas.

# 4 Chapman-Enskog theory for the GHD-Boltzmann equation

We apply now the ChE method to the GHD-Boltzmann equation. The procedure parallels the classical method presented in the previous section, which can be summarized as a two-step scheme:

1. Write suitable expansion of the state $\rho_p$ in parameters which are small in hydrodynamic limit. Assume that time dependence of the state is given by the time-dependence of the conserved hydrodynamic fields, which are not expanded in the small parameter of the theory. Expand GHD-Boltzmann equation and solve it order-by order finding corrections to the stationary state obtained at the leading order.

2. Evaluate corresponding corrections to the hydrodynamic pressure $\mathcal{P}$ and heat current $\mathcal{J}$. These after plugging into (57), yield hydrodynamic equations at subsequent orders (Euler scale, Navier-Stokes scale and so on) and provide expressions for transport coefficients.

However, there are two main differences with respect to the previous section: a) the thermodynamics of the underlying theory is not of a free system, b) there are two expansion parameters. In the general discussion presented in this section the second point will be especially important. More precisely as we will see the presence of the two expansion parameters leads to two types of contributions to the transport coefficients. Instead, the thermodynamics of the system will be important for the properties of the transport coefficients. Ultimately, it is the thermodynamics that is responsible for non-zero bulk viscosity unlike in the ideal gas. This we discuss in further sections.

## 4.1 General structure

Recall that the central assumption of the ChE method is that the time dependence of $\rho_p$ enters through the values of hydrodynamic fields, which we again collectively denote as $\{\varrho_a(x,t)\}$.

This then leads to the NS equations formulated in terms of these fields. However, equivalently we can assume that the time dependence of $\rho_{\mathrm{p}}$ is governed by the expectation values of the conserved charges $\{q_a(x, t)\}$. The resulting NS equations will be then automatically formulated in terms of them, or by using charge and current susceptibility matrices in terms of the corresponding chemical potentials. We will mainly use the second option as it provides a more symmetric presentation of the problem but occasionally we will also display results for the hydrodynamic fields as they are more standard.

We choose local conserved charges to control the time-dependence of $\rho_{\mathrm{p}}$,

$$\rho_{\mathrm{p}}(\lambda; x, t) \to \rho_{\mathrm{p}}(\lambda, x | \{q_a(x, t)\}), \tag{90}$$

with $\rho_{\mathrm{p}}(\lambda, x | \{q_a(x, t)\})$ called *normal solutions*. Their time evolution is given by

$$\partial_t \rho_{\mathrm{p}} = \sum_{a=0,1,2} \frac{\partial \rho_{\mathrm{p}}}{\partial q_a} \partial_t q_a = \sum_{a=0,1,2} \frac{\partial \rho_{\mathrm{p}}}{\partial q_a} G_a(\{q_b\}), \tag{91}$$

where we defined (below $a, b = 0, 1, 2$)

$$G_a(\{q_b\}) = -\partial_x j_a(\{q_b\}) + \frac{\delta_{\mathfrak{D}}}{2} \partial_x \int d\lambda \, \hbar_a(\lambda) \big(\mathfrak{D} \partial_x \rho_{\mathrm{p}}\big)(x, \lambda | \{q_b\}), \tag{92}$$

in accordance with the time evolution of the conserved charges (31). $G_a(\{q_b\})$ are nonlinear functionals of $\rho_{\mathrm{p}}$ and determine the dynamics of the hydrodynamic fields

$$\partial_t q_a = G_a(\{q_b\}). \tag{93}$$

The GHD-Boltzmann equation for the normal solutions becomes

$$\sum_{a=0,1,2} \frac{\partial \rho_{\mathrm{p}}}{\partial q_a} G_a(q_b) + \partial_x \big(v\rho_{\mathrm{p}}\big) = \frac{\delta_{\mathfrak{D}}}{2} \partial_x \big(\mathfrak{D} \partial_x \rho_{\mathrm{p}}\big) + \frac{1}{\delta_{\mathcal{I}}} \mathcal{I}[\rho_{\mathrm{p}}]. \tag{94}$$

This is an ordinary differential equation in $x$ in which time $t$ plays a role of a parameter. We assume that the normal solutions can be expanded as follows

$$\rho_{\mathrm{p}} = \sum_{n=0}^{\infty} \sum_{m=0}^{\infty} \delta_{\mathfrak{D}}^n \delta_{\mathcal{I}}^m \rho_{\mathrm{p}}^{(n,m)}, \tag{95}$$

where $\rho_{\mathrm{p}}^{(n,m)} \equiv \rho_{\mathrm{p}}^{(n,m)}(x, \lambda | \{\varrho_\alpha(x, t, \lambda)\})$. To make closer connection with the standard expansion we reparametrize the corrections as

$$\rho_{\mathrm{p}}^{(n,m)} = -\mathcal{C}^{(0,0)} \epsilon_0^{(n,m)}, \tag{96}$$

where $\mathcal{C}^{(0,0)}$ is the kernel $\mathcal{C}$ evaluated on $\rho_{\mathrm{p}}^{(0,0)}$. The reason for such change is the following. We would like to think of small corrections as perturbations to the bare pseudoenergy of the state, a quantity which has a natural expansion in ultra-local basis. Note that in the classical case the corrections are in fact of the form $-\rho^{(0)} \rho^{(n)}$, $n \geq 1$ which agree with our definition after realizing that for classical non-interacting gas $\mathcal{C}^{(0)} = \rho^{(0)}$. In what follows, we will drop the superscript from matrix $\mathcal{C}$.

A crucial point of the method is that we do not expand $q_a$ in $\delta_{\mathfrak{D}}, \delta_{\mathcal{I}}$. This implies that local values of the conserved charges are determined solely from $\rho_{\mathrm{p}}^{(0,0)}$,

$$q_a(x, t) = \int d\lambda \, \hbar_a(\lambda) \rho_{\mathrm{p}}^{(0,0)}(x, \lambda | \{\varrho_b\}),$$

$$0 = \int d\lambda \, \hbar_a(\lambda) \big(\mathcal{C} \epsilon_0^{(n,m)}\big)(x, \lambda | \{\varrho_b\}) = (\hbar_a | \epsilon_0^{(n,m)}), \qquad (n, m) \neq (0, 0). \tag{97}$$

Which once again means that corrections $\epsilon_0^{(n,m)}$ are orthogonal to collision invariants in the sense of hydrodynamic scalar product. These equations form constraints (constitutive conditions) on the expansion of the normal solutions. The evolution operator for hydrodynamic fields is expanded as well

$$G_a(\{\mathcal{q}_b\}) = \sum_{n=0}^{\infty}\sum_{m=0}^{\infty} \delta_{\mathfrak{D}}^n \delta_{\mathcal{I}}^m G_a^{(n,m)}(\{\mathcal{q}_b\}). \tag{98}$$

The first equation (level 1 ChE approximation) in the hierarchy is

$$\mathcal{I}[\rho_{\mathrm{p}}^{(0,0)}] = 0, \tag{99}$$

which implies that $\rho_{\mathrm{p}}^{(0,0)}$ is a stationary state of the collision integral, i.e. thermal boosted state. The next ones, at level 2, are

$$\Gamma\epsilon_0^{(1,0)} = 0, \qquad \sum_{a=0,1,2} \frac{\partial\rho_{\mathrm{p}}^{(0,0)}}{\partial\mathcal{q}_a} G_a^{(0,0)}(\{\mathcal{q}_b\}) + \partial_x\left(v^{(0,0)}\rho_{\mathrm{p}}^{(0,0)}\right) = \Gamma\epsilon_0^{(0,1)}, \tag{100}$$

we defined here $\Gamma = -\mathcal{I}^{(1)}\mathcal{C}$ where $\mathcal{I}^{(1)}$ is the functional derivative of $\mathcal{I}[\rho_{\mathrm{p}}]$ evaluated at $\rho_{\mathrm{p}} = \rho_{\mathrm{p}}^{(0,0)}$. The first equation is solved by any function from the kernel of operator $\Gamma$. Its kernel contains collision invariants $\hbar_a$ with $a = 0, 1, 2$ and therefore any solution is of the form $\epsilon_0^{(1,0)} = \sum_{a=0}^2 c_a(x,t)\hbar_a(\lambda)$. However, according to the constitutive conditions (97) only the leading term in the ChE expansion contributes to the hydrodynamic fields. This enforces that $\epsilon_0^{(1,0)} = 0$. Alternatively, equation $\Gamma\epsilon_0^{(1,0)} = 0$ can be understood as coming from expanding

$$\mathcal{I}\left[\rho_{\mathrm{p}}^{(0,0)} - \delta_{\mathfrak{D}}\mathcal{C}\epsilon_0^{(1,0)}\right] = 0. \tag{101}$$

On the other hand, this equation itself just states that the combination $\rho_{\mathrm{p}}^{(0,0)} - \delta_{\mathfrak{D}}\mathcal{C}\epsilon_0^{(1,0)}$ is a stationary state. Hence, choosing $\epsilon_0^{(1,0)} \neq 0$ has an effect of shifting the stationary, thermal boosted state and therefore is redundant. The role of the constitutive conditions is to remove this redundancy.

The second equation of the ChE hierarchy at level 2 given in (100) determines $\epsilon_0^{(0,1)}$. Note that it does not involve explicitly the diffusion term of the GHD and is structurally similar to the standard ChE equation at level 2, c.f. (82).

As mentioned earlier, hydrodynamic equations at subsequent orders follow from evaluating corrections to pressure $\mathcal{P}$ and heat current $\mathcal{J}$. The same equations in $\{\mathcal{q}_0, \mathcal{q}_1, \mathcal{q}_2\}$ or $\{\beta_0, \beta_1, \beta_2\}$ variables follow now from evaluating $G_a(\{\mathcal{q}_b\})$ on the series expansion of $\rho_{\mathrm{p}}$. To discuss a general structure of the resulting equations it is useful to decompose $G_a$ in two contributions differing in the scaling parameters,

$$\begin{aligned} G_{a,v}(\{\mathcal{q}_b\}) &= -\int d\lambda\, \hbar_a(\lambda)\partial_x\left(v\rho_{\mathrm{p}}(x,\lambda|\{\mathcal{q}_b\})\right) = \sum_{n=0}^{\infty}\sum_{m=0}^{\infty} \delta_{\mathfrak{D}}^n \delta_{\mathcal{I}}^m G_{a,v}^{(n,m)}(\{\mathcal{q}_b\}), \\ G_{a,\mathfrak{D}}(\{\mathcal{q}_b\}) &= \frac{\delta_{\mathfrak{D}}}{2}\int d\lambda\, \hbar_a(\lambda)\partial_x\left(\mathfrak{D}\partial_x\rho_{\mathrm{p}}(x,\lambda|\{\mathcal{q}_b\})\right) = \sum_{n=1}^{\infty}\sum_{m=0}^{\infty} \delta_{\mathfrak{D}}^n \delta_{\mathcal{I}}^m G_{a,\mathfrak{D}}^{(n,m)}(\{\mathcal{q}_b\}). \end{aligned} \tag{102}$$

The hydrodynamic equations, up to and including the first order terms, are then

$$\partial_t\mathcal{q}_a = \underbrace{G_{a,v}^{(0,0)}[\rho_{\mathrm{p}}^{(0,0)}]}_{\text{Euler}} + \underbrace{\delta_{\mathfrak{D}}G_{a,\mathfrak{D}}^{(1,0)}[\rho_{\mathrm{p}}^{(0,0)}] + \delta_{\mathcal{I}}G_{a,v}^{(0,1)}[\epsilon_0^{(0,1)}]}_{\text{NS}}, \tag{103}$$

where in the brackets $[\cdot]$ we have indicated dependence on correction to the state (95).

In the following sections we will derive the Euler and NS hydrodynamics equations from the ChE equation. The two relevant functions are $\rho_{\mathrm{p}}^{(0,0)}$ and $\epsilon_0^{(0,1)}$ because $\epsilon^{(1,0)} = 0$. Therefore, to simplify the notation, we write $\rho_{\mathrm{p}}^{(0)} \equiv \rho_{\mathrm{p}}^{(0,0)}$, $\epsilon_0^{(1)} \equiv \epsilon_0^{(0,1)}$ and $G_a^{(0)} \equiv G_a^{(0,0)}$. The ChE equations are then

$$\mathcal{I}[\rho_{\mathrm{p}}^{(0)}] = 0, \qquad \sum_{a=0,1,2} \frac{\partial \rho_{\mathrm{p}}^{(0)}}{\partial \mathscr{q}_a} G_a^{(0)}(\mathscr{q}_b) + \partial_x\left(v^{(0)} \rho_{\mathrm{p}}^{(0)}\right) = \Gamma \epsilon_0^{(1)}. \qquad (104)$$

It is worth emphasising again that the GHD diffusion enters these equations only through $G_a(\{\mathscr{q}_b\})$. In the following two sections we discuss the equations (104) and (103) at the Euler and at the NS scales respectively.

There is also an alternative route to the NS equations which capitalizes on the knowledge of their form. Therefore, instead of computing $G_a$ and rediscovering the NS equations, we can directly compute the hydrodynamic pressure $\mathcal{P}$ and heat current $\mathcal{J}$. Both quantities have an expansion in $\delta$'s

$$\mathcal{P} = \sum_{n=0}^{\infty} \sum_{m=0}^{\infty} \delta_{\mathfrak{D}}^n \delta_{\mathcal{I}}^m \mathcal{P}^{(n,m)}, \qquad \mathcal{J} = \sum_{n=0}^{\infty} \sum_{m=0}^{\infty} \delta_{\mathfrak{D}}^n \delta_{\mathcal{I}}^m \mathcal{J}^{(n,m)}, \qquad (105)$$

and knowing the expansion of the normal solutions they can be evaluated order by order.

This offers a shortcut in deriving the hydrodynamics and occasionally we will take advantage of it. Note however, that obtaining the hydrodynamics by computing $G_a$ is more systematic as it does not anticipate the structure of the resulting equations. The later, in the present context, is dictated by the Galilean invariance. This is specific to continuum non-relativistic systems and does not hold for example in the spin chains systems or relativistic theories.

## 4.2 Euler scale

Consider first of the equations in (104). This equation implies that $\rho_{\mathrm{p}}^{(0)}$ is a stationary state of the collision integral. Because the collision integral acts only on the rapidity dependence of $\rho_{\mathrm{p}}^{(0)}$, the parameters of the stationary state may vary in space and time. This implies that a stationary state of the collision integral is a boosted thermal state that can be characterized by 3 generalized temperatures $\{\beta_0, \beta_1, \beta_2\}$ or equivalently by hydrodynamic fields $\{\varrho, u, T\}$. With this, we can evaluate dynamic pressure and heat current at the leading order

$$\mathcal{P}^{(0,0)} = \mathcal{P}_v^{(0,0)} = P, \qquad \mathcal{J}^{(0,0)} = \mathcal{J}_v^{(0,0)} = 0, \qquad (106)$$

where $P$ is the thermodynamic pressure from TBA. Computations involve some operations on TBA objects and are demonstrated in Supplemental Material of [51]. We note that once again that diffusion contributions do not enter at zeroth order due to $\delta_{\mathfrak{D}}$ prefactor in (64). Plugging this into (57) we recover the standard Euler hydrodynamics

$$\partial_t \varrho = -\partial_x(\varrho u), \qquad \partial_t(\varrho u) = -\partial_x\left(\varrho u^2 + P\right), \qquad \partial_t(\varrho e) = -\partial_x(\varrho u e) - P \partial_x u. \qquad (107)$$

We note now that the same equations can be derived in $\{\beta_0, \beta_1, \beta_2\}$ variables by finding operators $G_a^0(\{\mathscr{q}_b\})$ and expressing them in these variables using (27),

$$G_a^{(0)}(\{\mathscr{q}_b\}) = -\hbar_a \partial_x\left(v^{(0)} \rho_{\mathrm{p}}^{(0)}\right) = \mathcal{B}_{ab} \partial_x \beta_b, \qquad (108)$$

where we used eq. (27). This leads to the following representation of the Euler scale equations

$$\sum_{b=0}^{2} (\mathcal{C}_{ab} \partial_t \beta_b + \mathcal{B}_{ab} \partial_x \beta_b) = 0, \qquad a = 0, 1, 2, \qquad (109)$$

which will be useful in the next section. We note that these equations have appeared previously in [79].

### 4.3 Navier-Stokes scale

At the NS scale the hydrodynamic equations receive two types of corrections. The first one is due to the GHD diffusion and follow from evaluating the diffusion operator on the stationary state $\rho_{\mathrm{p}}^{(0)}$. The second type of corrections come from the collision integral and require solving the second ChE equation. We start with the former. In terms of the corrections to pressure and heat current we have

$$\mathcal{P}_{\mathfrak{D}}^{(1,0)} = -\hbar_1 \mathfrak{D}\mathcal{C}\partial_x \rho_{\mathrm{p}}^{(0)}/2, \qquad u\mathcal{P}_{\mathfrak{D}}^{(1,0)} + \mathcal{J}_{\mathfrak{D}}^{(1,0)} = -\hbar_2 \mathfrak{D}\mathcal{C}\partial_x \rho_{\mathrm{p}}^{(0)}/2, \tag{110}$$

moreover, contributions to $G_a$ at this level are

$$G_{a,\mathfrak{D}}^{(1,0)} = \frac{1}{2}\partial_x \left( \hbar_a \mathfrak{D}\partial_x \rho_{\mathrm{p}}^{(0)} \right). \tag{111}$$

In both cases we evaluate the coefficients by transferring the spatial derivative from the distribution to the generalized temperatures using (27)

$$\partial_x \rho_{\mathrm{p}}^{(0)} = -\mathcal{C}\sum_{b=0}^{2} \hbar_b \partial_x \beta_b. \tag{112}$$

This reduces the problem to determining $\hbar_a \mathfrak{D}\mathcal{C}\hbar_b$ which in principle leads to 9 coefficients. However, the diffusion operator has left zero eigenvector $\hbar_0$ and right zero eigenvector $\mathcal{C}\hbar_0$. This simplifies the problem from determining 9 to 4 coefficients. Furthermore, from the Galilean invariance it follows that in fact there are only 2 independent coefficients. To demonstrate this we use the boost operators to write

$$\hbar_a \mathfrak{D}\mathcal{C}\hbar_b = \hbar_a b_u^{-1}\bar{\mathfrak{D}}\bar{\mathcal{C}}b_u \hbar_b, \tag{113}$$

from which we find

$$\begin{aligned} \hbar_1 \mathfrak{D}\mathcal{C}\hbar_1 &= \hbar_1 \bar{\mathfrak{D}}\bar{\mathcal{C}}\hbar_1, & \hbar_1 \mathfrak{D}\mathcal{C}\hbar_2 &= u\hbar_1 \bar{\mathfrak{D}}\bar{\mathcal{C}}\hbar_1, \\ \hbar_2 \mathfrak{D}\mathcal{C}\hbar_1 &= u\hbar_1 \bar{\mathfrak{D}}\bar{\mathcal{C}}\hbar_1, & \hbar_2 \mathfrak{D}\mathcal{C}\hbar_2 &= \hbar_2 \bar{\mathfrak{D}}\bar{\mathcal{C}}\hbar_2 + u^2 \hbar_1 \bar{\mathfrak{D}}\bar{\mathcal{C}}\hbar_1, \end{aligned} \tag{114}$$

which indeed involve only 2 independent coefficients $\hbar_1 \bar{\mathfrak{D}}\bar{\mathcal{C}}\hbar_1$ and $\hbar_2 \bar{\mathfrak{D}}\bar{\mathcal{C}}\hbar_2$ which depend on the local density $\varrho(x,t)$ and local temperature $T(x,t)$ characterizing the local thermal state $\rho_{\mathrm{p}}^{(0)}$.

For the dynamic pressure and the heat current we get

$$\mathcal{P}_{\mathfrak{D}}^{(1,0)} = -\zeta_{\mathfrak{D}}\partial_x u, \qquad \mathcal{J}_{\mathfrak{D}}^{(1,0)} = -\kappa_{\mathfrak{D}}\partial_x T, \tag{115}$$

with the corresponding transport coefficients

$$\zeta_{\mathfrak{D}} = \hbar_1 \bar{\mathfrak{D}}\bar{\mathcal{C}}\hbar_1/(2T), \qquad \kappa_{\mathfrak{D}} = \hbar_2 \bar{\mathfrak{D}}\bar{\mathcal{C}}\hbar_2/(2T^2). \tag{116}$$

Moreover, expressing the $G$ operators through the two independent coefficients $\hbar_1 \bar{\mathfrak{D}}\bar{\mathcal{C}}\hbar_1$ and $\hbar_2 \bar{\mathfrak{D}}\bar{\mathcal{C}}\hbar_2$, we find

$$\begin{aligned} G_{0,\mathfrak{D}}^{(1,0)}[\rho_{\mathrm{p}}^{(0)}] &= 0, \\ G_{1,\mathfrak{D}}^{(1,0)}[\rho_{\mathrm{p}}^{(0)}] &= \frac{1}{2}\partial_x \left( \hbar_1 \bar{\mathfrak{D}}\bar{\mathcal{C}}\hbar_1 \beta_2 \partial_x u \right), \\ G_{2,\mathfrak{D}}^{(1,0)}[\rho_{\mathrm{p}}^{(0)}] &= -\frac{1}{2}\partial_x \left( \hbar_2 \bar{\mathfrak{D}}\bar{\mathcal{C}}\hbar_2 \partial_x \beta_2 \right) + \frac{1}{2}\partial_x \left( u\hbar_1 \bar{\mathfrak{D}}\bar{\mathcal{C}}\hbar_1 \beta_2 \partial_x u \right). \end{aligned} \tag{117}$$

In writing this expressions we have used that (it follows from (42))

$$\partial_x \beta_1 + u \partial_x \beta_2 = -\beta_2 \partial_x u, \tag{118}$$

which points to using $u = -\beta_1/\beta_2$ as a hydrodynamic field instead of $\beta_1$.

The knowledge of $G_{a,\mathfrak{D}}^{(1,0)}[\rho_{\mathrm{p}}^{(0)}]$ allows us to write down the equations for the hydrodynamic fields which beside the Euler scale include also the effect of the GHD diffusion. As we shall see, the structure of the contributions to $G_{a,\mathfrak{D}}^{(1,0)}[\rho_{\mathrm{p}}^{(0)}]$ and $G_{a,\nu}^{(1)}[\epsilon_0^{(1)}]$ is the same. Therefore, we will discuss the resulting equations after computing the contribution from the collision term. This requires solving the ChE equation which we discuss next.

We turn now our attention to the second equation of (104) which determines $\epsilon_0^{(1)}$ and in turn determines $G_{a,\nu}^{(1)}[\epsilon_0^{(1)}]$. The equation is

$$\partial_t \rho_{\mathrm{p}}^{(0)} + \partial_x \left(v^{(0)} \rho_{\mathrm{p}}^{(0)}\right) = \Gamma \epsilon_0^{(1)}, \tag{119}$$

and

$$G_{a,\nu}^{(1)}[\epsilon_0^{(1)}] = \partial_x \left(\hbar_a \mathcal{B} \epsilon_0^{(1)}\right). \tag{120}$$

Recall that since $\rho_{\mathrm{p}}^{(0)}$ is a thermal boosted state it is completely characterized by fields $\{\varrho_\beta\}$ or equivalently by the three chemical potentials $\beta_a$ with $a = 0, 1, 2$. Equation (119) determines the correction to pseudoenergy $\epsilon_0^{(1)}$ which additionally has to fulfill the constitutive relations (97) which can be written as

$$0 = \hbar_a \mathcal{C} \epsilon_0^{(1)}, \qquad a = 0, 1, 2. \tag{121}$$

We observe that the constitutive relation itself is enough to conclude that $G_{0,\nu}^{(1)}[\epsilon_0^{(1)}] = 0$. Indeed, we have

$$G_{0,\nu}^{(1)}[\epsilon_0^{(1)}] = \partial_x \left(\hbar_0 \mathcal{B} \epsilon_0^{(1)}\right) = \partial_x \left(\hbar_1 \mathcal{C} \epsilon_0^{(1)}\right) = 0. \tag{122}$$

Together with the first formula in (117) this implies that there is no diffusion of the density as expected in a single component fluid.

We now go back to the full problem (119) and start by expressing the left hand side by the derivatives of the chemical potentials

$$\partial_t \rho_{\mathrm{p}}^{(0)} + \partial_x (v^{(0)} \rho_{\mathrm{p}}^{(0)}) = -\sum_{b=0}^{2} (\mathcal{C} \hbar_b \partial_t \beta_b + \mathcal{B} \hbar_b \partial_x \beta_b), \tag{123}$$

where we used (27). The chemical potentials evolve now according to the Euler scale equations (109).

The Euler scale hydrodynamics is a truncation of this set to 3 equations for $a = 0, 1, 2$. Therefore, this expression in general is non-zero and gives rise to finite $\epsilon_0^{(1)}$. Denote by $\mathcal{C}^{\mathrm{Eul}}$ and $\mathcal{B}^{\mathrm{Eul}}$ reductions of the susceptibility matrices $\mathcal{C}_{ab}$ and $\mathcal{B}_{ab}$ to a subspace of $a, b = 0, 1, 2$. The Euler scale hydrodynamics (109) gives

$$\partial_t \beta_a = -\sum_{b=0}^{2} \mathcal{A}_{ab}^{\mathrm{Eul}} \partial_x \beta_b, \tag{124}$$

where $\mathcal{A}^{\mathrm{Eul}} = \left(\mathcal{C}^{\mathrm{Eul}}\right)^{-1} \mathcal{B}^{\mathrm{Eul}}$ (invertibility of $\mathcal{C}^{\mathrm{Eul}}$ is demonstrated in Sec. 2.3) and

$$\partial_t \rho_{\mathrm{p}}^{(0)} + \partial_x (v^{(0)} \rho_{\mathrm{p}}^{(0)}) = \sum_{b,c=0}^{2} \left(\delta_{bc} \mathcal{B} \hbar_c - \mathcal{C} \hbar_b \mathcal{A}_{bc}^{\mathrm{Eul}}\right) \partial_x \beta_c. \tag{125}$$

This gives then the following equation for $\epsilon_0^{(1)}$,

$$\sum_{b,c=0}^{2} \left( \delta_{bc} \mathcal{B} \hbar_c - \mathcal{C} \hbar_b \mathcal{A}_{bc}^{\mathrm{Eul}} \right) \partial_x \beta_c = \Gamma \epsilon_0^{(1)} . \tag{126}$$

The rapidity variable on the left hand side appears only in the terms in front of the spatial derivative. Therefore, the space-time and rapidity dependence factorize and the solution has to be of the following form

$$\epsilon_0^{(1)}(\lambda; x, t) = -\sum_{c=0}^{2} w_c(\lambda) \partial_x \beta_c(x, t) . \tag{127}$$

Substituting this expression to equation (126) we find that functions $w_c$ obey

$$\Gamma w_c = \mathcal{B} \hbar_c - \sum_{b=0}^{2} \mathcal{C} \hbar_b \mathcal{A}_{bc}^{\mathrm{Eul}} , \qquad c = 0, 1, 2 . \tag{128}$$

Additionally, function $\epsilon_0^{(1)}$ obeys the constitutive relations (121) which translate into the following conditions on functions $w_c$,

$$\hbar_a \mathcal{C} w_c = 0 , \qquad a, c = 0, 1, 2 . \tag{129}$$

Equations (128) and (129) are the final equations determining the structure of $\epsilon_0^{(1)}$. They depend on the local local thermal boosted state $\rho_{\mathrm{p}}^{(0)}(\lambda; x, t)$. This in turn is characterized by the density $\varrho$, temperature $T$ and boost $u$. From the Galilean invariance of the theory it follows that the collision integral is an invariant object. Therefore, by going to the reference frame in which fluid-cell at position $(x, t)$ is at rest the equations are invariant,

$$\bar{\Gamma} \bar{w}_c = \bar{\mathcal{B}} \hbar_c - \sum_{b=0}^{2} \bar{\mathcal{C}} \hbar_b \bar{\mathcal{A}}_{bc}^{\mathrm{Eul}} , \qquad \hbar_a \bar{\mathcal{C}} \bar{w}_c = 0 , \qquad a, c = 0, 1, 2 , \tag{130}$$

where all the formulas are now evaluated for a thermal state with density $\varrho$ and temperature $T$. The transformation rules for $\Gamma$ and $w_c$ are $\Gamma = b_u^{-1} \bar{\Gamma} b_u$ and $w_c = b_u^{-1} \bar{w}_d (g^T(u))_{dc}$ with a summation over $d$ implied. The details are presented in Appendix A.2.

In a thermal state matrix $\bar{\mathcal{A}}^{\mathrm{Eul}}$ has only four non-zero matrix-elements, c.f. (43),

$$\bar{\mathcal{A}}^{\mathrm{Eul}} = \begin{pmatrix} 0 & \bar{\mathcal{A}}_{01}^{\mathrm{Eul}} & 0 \\ \bar{\mathcal{A}}_{10}^{\mathrm{Eul}} & 0 & \bar{\mathcal{A}}_{12}^{\mathrm{Eul}} \\ 0 & \bar{\mathcal{A}}_{21}^{\mathrm{Eul}} & 0 \end{pmatrix} = \begin{pmatrix} 0 & \frac{\bar{\mathcal{B}}_{12} \bar{\mathcal{C}}_{02} - \bar{\mathcal{B}}_{01} \bar{\mathcal{C}}_{22}}{\bar{\mathcal{C}}_{02}^2 - \bar{\mathcal{C}}_{00} \bar{\mathcal{C}}_{22}} & 0 \\ \frac{\bar{\mathcal{B}}_{01}}{\bar{\mathcal{C}}_{11}} & 0 & \frac{\bar{\mathcal{B}}_{12}}{\bar{\mathcal{C}}_{11}} \\ 0 & \frac{\bar{\mathcal{B}}_{12} \bar{\mathcal{C}}_{00} - \bar{\mathcal{B}}_{01} \bar{\mathcal{C}}_{02}}{\bar{\mathcal{C}}_{00} \bar{\mathcal{C}}_{22} - \bar{\mathcal{C}}_{02}^2} & 0 \end{pmatrix} . \tag{131}$$

This simplifies the structure of the equations (130). Writing them explicitly we find

$$\bar{\Gamma} \bar{w}_a = \bar{\mathcal{B}} \hbar_a - \bar{\mathcal{C}} \hbar_1 \bar{\mathcal{A}}_{1a}^{\mathrm{Eul}} , \qquad a = 0, 2 , \tag{132}$$

$$\bar{\Gamma} \bar{w}_1 = \bar{\mathcal{B}} \hbar_1 - \bar{\mathcal{C}} \left( \hbar_0 \bar{\mathcal{A}}_{01}^{\mathrm{Eul}} + \hbar_2 \bar{\mathcal{A}}_{21}^{\mathrm{Eul}} \right) . \tag{133}$$

Recall again that in the Galilean invariant systems $\mathcal{B} \hbar_0 = \mathcal{C} \hbar_1$. This implies that $\bar{\mathcal{B}}_{01} = \bar{\mathcal{C}}_{11}$ and $\bar{\mathcal{A}}_{10}^{\mathrm{Eul}} = 1$. In the consequence $\bar{w}_0 = 0$. This reduces the problem to solving for two unknown functions. Notice also that $\bar{w}_1$ is an even function of the rapidity while $\bar{w}_2$ is odd.

Boosting the state in general mixes different contributions according to

$$w_a = b_u^{-1} \sum_{b=0}^{2} \bar{w}_b g(u)_{ab} . \tag{134}$$

Thanks to the triangular structure of the transformation matrix and $\bar{w}_0 = 0$, the result is simple

$$w_0 = 0, \qquad w_1 = b_u^{-1}\bar{w}_1, \qquad w_2 = b_u^{-1}(\bar{w}_2 + u\bar{w}_1), \tag{135}$$

or more explicitly

$$w_0(\lambda) = 0, \qquad w_1(\lambda) = \bar{w}_1(\lambda - u), \qquad w_2(\lambda) = (\bar{w}_2(\lambda - u) + u\bar{w}_1(\lambda - u)). \tag{136}$$

We evaluate now the $G_{a,\nu}^{(1)}$ operators. As mentioned earlier $G_{0,\nu}^{(1)} = 0$. The other two evaluate as follows

$$G_{1,\nu}^{(1)} = -(\hbar_1 \mathcal{B} w_1 \partial_x \beta_1 + \hbar_1 \mathcal{B} w_2 \partial_x \beta_2), \qquad G_{2,\nu}^{(1)} = -(\hbar_2 \mathcal{B} w_1 \partial_x \beta_1 + \hbar_2 \mathcal{B} w_2 \partial_x \beta_2). \tag{137}$$

We write now

$$\hbar_a \mathcal{B} w_c = \hbar_a b_u^{-1} (\bar{\mathcal{B}} + u\bar{\mathcal{C}}) b_u w_c = \hbar_a b_u^{-1} \bar{\mathcal{B}} b_u w_c, \tag{138}$$

where we used the constitutive relations. More explicitly, we find

$$\begin{aligned}
\hbar_1 \mathcal{B} w_1 &= \hbar_1 \bar{\mathcal{B}} \bar{w}_1, & \hbar_1 \mathcal{B} w_2 &= u\hbar_1 \bar{\mathcal{B}} \bar{w}_1, \\
\hbar_2 \mathcal{B} w_1 &= u\hbar_1 \bar{\mathcal{B}} \bar{w}_1, & \hbar_2 \mathcal{B} w_2 &= \hbar_2 \bar{\mathcal{B}} \bar{w}_2 + u^2 \hbar_1 \bar{\mathcal{B}} \bar{w}_1,
\end{aligned} \tag{139}$$

which is of the same structure as (114) and therefore the $G$ operators takes also form analogous to (117).

This allows us to write the NS equations in $\{\beta_0, \beta_1, \beta_2\}$ variables

$$\sum_{b=0}^{2} \left( \mathcal{C}_{ab} \partial_t \beta_b + \mathcal{B}_{ab} \partial_x \beta_b + \frac{1}{2} \partial_x (D_{ab}^{\text{NS}} \partial_x \beta_b) \right) = 0, \qquad a = 0, 1, 2, \tag{140}$$

where the diffusion coefficients are $D^{\text{NS}} = \mathfrak{D}\mathcal{C} + D^{\text{coll}}$ with

$$(\mathfrak{D}\mathcal{C})_{ab} = \hbar_a \mathfrak{D}\mathcal{C} \hbar_b, \qquad D_{ab}^{\text{coll}} = 2\hbar_a \mathcal{B} w_b, \tag{141}$$

expressed through in total 4 transport coefficients $\hbar_1 \bar{\mathfrak{D}}\bar{\mathcal{C}} \hbar_1$, $\hbar_2 \bar{\mathfrak{D}}\bar{\mathcal{C}} \hbar_2$, $\hbar_1 \bar{\mathcal{B}} \bar{w}_1$ and $\hbar_2 \bar{\mathcal{B}} \bar{w}_2$. In the Galilean invariant systems $D_{0b}^{\text{NS}} = D_{a0}^{\text{NS}} = 0$.

Instead, computing corrections to pressure (55) and heat current (56) we find

$$\mathcal{P}_\nu^{(1,0)} = -\zeta_{\mathcal{I}} \partial_x u, \qquad \mathcal{J}_\nu^{(1,0)} = -\kappa_{\mathcal{I}} \partial_x T, \tag{142}$$

with

$$\zeta_{\mathcal{I}} = \hbar_1 \bar{\mathcal{B}} \bar{w}_1 / T, \qquad \kappa_{\mathcal{I}} = \hbar_2 \bar{\mathcal{B}} \bar{w}_2 / T^2. \tag{143}$$

The Eq. (140) can be recast into the more canonical form (57). In order to do so, we apply (29) to the first two terms and rewrite transport coefficients $D^{NS}$ using (114) and (139). As a result we recover exactly (57) with $\mathcal{P}$ and $\mathcal{J}$ found in this section.

Finally, let us summarize the main results of the method. As described above, we have found four non-trivial transport coefficients $D^{\text{NS}}$ which enter the NS equations written for the dynamics of $\{\beta_0, \beta_1, \beta_2\}$. At the same time, we have recovered Newton's and Fourier laws, which appear when NS equations are expressed with hydrodynamic fields. Contributions to transport coefficients are computed straightforwardly from (116). On the other hand, contributions to the transport coefficients from the collision integral (143) are found from solution of integral equations (132). Note that these equations are slightly different than the equations given in the main text of [51]. The difference lies in using the orthonormal (with respect to hydrodynamic inner product (26)) basis $\{h_n\}_{n \in \mathbb{N}}$ in our earlier work as opposed to the ultralocal basis $\{\hbar_n\}_{n \in \mathbb{N}}$ in the present paper. The equivalence between the two representations is demonstrated in Appendix A.3.

## 4.4 External potential

We generalize now our results to systems with external potential $U(x)$ that couples to the density of particles. The GHD-Boltzman equation is

$$\partial_t \rho_{\mathrm{p}} + \partial_x \left( v \rho_{\mathrm{p}} \right) = \frac{1}{2} \partial_x \left( \mathfrak{D} \partial_x \rho_{\mathrm{p}} \right) + \mathcal{I}[\rho_{\mathrm{p}}] + (\partial_x U) \partial_\lambda \rho_{\mathrm{p}} . \tag{144}$$

Following the same procedure as before, we start by deriving the mesoscopic conservation laws. They take the expected form, see eq. (6),

$$\partial_t \mathcal{q}_0 + \partial_x \mathcal{q}_1 = 0 ,$$
$$\partial_t \mathcal{q}_1 + \partial_x \left( \frac{\mathcal{q}_1^2}{\mathcal{q}_0} + \mathcal{P} \right) = -(\partial_x U) \mathcal{q}_0 , \tag{145}$$
$$\partial_t \mathcal{q}_2 + \partial_x \left( \frac{\mathcal{q}_1 \mathcal{q}_2}{\mathcal{q}_0} + u\mathcal{P} + \mathcal{J} \right) = -(\partial_x U) \mathcal{q}_1 ,$$

with $\mathcal{P}$ and $\mathcal{J}$ of the same form as discussed in Section 2.5.

The external potential in principle introduces a new length-scale to the problem. However, at the level of the conservation laws its effect enters only through the hydrodynamic fields. Because of that it can be treated exactly within the ChE method which we now demonstrate.

The GHD equation for the normal solutions takes the following form

$$\sum_{a=0,1,2} \frac{\partial \rho_{\mathrm{p}}}{\partial \mathcal{q}_a} G_a(\mathcal{q}_b) + \partial_x \left( v \rho_{\mathrm{p}} \right) = \frac{1}{2} \partial_x \left( \mathfrak{D} \partial_x \rho_{\mathrm{p}} \right) + \mathcal{I}[\rho_{\mathrm{p}}] - \mathfrak{f} \partial_\lambda \rho_{\mathrm{p}} , \tag{146}$$

where $\mathfrak{f} = -\partial_x U$. Solution to this equation is a particle distribution function $\rho_{\mathrm{p}}^{\mathrm{ChE}}$ determined by the local values of the 3 conserved charges. From the point of view of the GGE this state can be described by the 3 lowest chemical potentials. We emphasize that the state is not a boosted thermal state and hence the higher chemical potentials are not zero. However, they are not free parameters either, they are functions of the 3 lowest chemical potentials by the virtue of eq. (146).

Because of these relations, the susceptibility matrix of state described by $\rho_{\mathrm{p}}^{\mathrm{ChE}}$ does not take the usual form. Indeed, the standard susceptibility matrix assumes that all the chemical potentials are free parameters. We define the ChE susceptibility matrix ($a, b = 0, 1, 2$)

$$\mathcal{C}_{ab}^{\mathrm{ChE}} = -\frac{\partial \mathcal{q}_a}{\partial \beta_b} , \tag{147}$$

where $\mathcal{q}_a$ are now values of conserved charges in state $\rho_{\mathrm{p}}^{\mathrm{ChE}}$. As with the GGE susceptibility matrix, we can use the ChE susceptibility matrix to evaluate various expressions. For example,

$$\partial_\lambda \rho_{\mathrm{p}}^{\mathrm{ChE}} = -\mathcal{C}^{\mathrm{ChE}} \hbar_b' \beta_b , \tag{148}$$

with the summation over the 3 independent chemical potentials. In a similar fashion we can evaluate the additional contribution to $G_a$ from the external potential

$$\mathfrak{f} \int \mathrm{d}\lambda \, \hbar_a(\lambda) \partial_\lambda \rho_{\mathrm{p}}(x, \lambda | \{\mathcal{q}_b\}) = -\mathfrak{f} \hbar_a \mathcal{C}^{\mathrm{ChE}} \hbar_b' \beta_b . \tag{149}$$

Collecting then the terms proportional to $\mathfrak{f}$ in eq. (146) we find

$$-\mathfrak{f} \left( \sum_{a=0}^{2} \frac{\partial \rho_{\mathrm{p}}}{\partial \mathcal{q}_a} \hbar_a + 1 \right) \mathcal{C}^{\mathrm{ChE}} \hbar_b' \beta_b , \tag{150}$$

where used the expression for derivative of the particle distribution with respect to the rapidity introduced above. On the other hand, assuming that the ChE susceptibility matrix is invertible, we have

$$\frac{\partial \rho_{\mathrm{p}}}{\partial \mathcal{q}_a} = \sum_{c=0}^{2} \frac{\partial \rho_{\mathrm{p}}}{\partial \beta_c} \frac{\partial \beta_c}{\partial \mathcal{q}_a} = -\mathcal{C}^{\mathrm{ChE}} \sum_{c=0}^{2} \hbar_c \left(\mathcal{C}_{ca}^{\mathrm{ChE}}\right)^{-1}, \tag{151}$$

and therefore

$$\sum_{a=0}^{2} \frac{\partial \rho_{\mathrm{p}}}{\partial \mathcal{q}_a} \hbar_a = -1. \tag{152}$$

Hence, the part proportional to $\mathfrak{f}$ drops from the equations for the normal solutions. Therefore, the external potential does not modify the transport coefficients of the NS equations.

# 5 Crossover between NS diffusion and KPZ superdiffusion

So far, we have shown the emergence of NS equations in nearly integrable quantum gases described by the Hamiltonian (4). However, transport in one-dimensional momentum conserving fluids is expected to be anomalous at the largest space-time scales [68–71], in contrast to diffusion predicted by NS hydrodynamics. Moreover, the so-called long-time tails [80] in autocorrelation functions are expected to occur, leading to the divergence of Green-Kubo formulas for transport coefficients, again as opposed to finite values found by us. On the other hand, there are numerical and analytical evidences that at intermediate space-time scale conventional NS hydrodynamics prevails [81–85]. In particular in [82], a classical system admitting a collision integral description was considered. At large length scales, anomalous transport was found, whereas for smaller scales the diffusive behavior with diffusion constants predicted by kinetic theory was confirmed.

In this section, we will address these issues. In particular, we will look at the behavior of the system across different length scales and uncover that while hydrodynamics indeed become anomalous at the largest length scales, there is a clear regime, where the NS description (with finite transport coefficients) is the correct hydrodynamic theory of the system. Existence of such a regime is essentially due to the fact, that we are close to the integrable model, which results in large transport coefficients, allowing diffusive terms to compete with anomalous contributions at finite, but small inverse length scales $k$. The general picture, which emerges from our analysis is presented in Fig.1. We will present the estimation of different crossover regions in what follows. To start our analysis, let us estimate on which characteristic length scale the NS equations are expected to emerge. The collision integral for a nearly integrable system follows from Fermi's Golden Rule and scales with the small integrability-breaking parameter $\lambda$ as

$$\mathcal{I} \sim \lambda^2, \tag{153}$$

leading to the scaling of the characteristic length scale $l_{\mathcal{I}}$ discussed in Sec. 2.6 as $l_{\mathcal{I}} \sim \lambda^{-2}$. The crossover from GHD to NS happens around the point, when the Knudsen number $\delta_{\mathcal{I}} = l_{\mathcal{I}}/l$ becomes of order one. This leads to crossover length scale

$$l_{\mathrm{NS/GHD}}^* \sim \lambda^{-2}. \tag{154}$$

The same estimate can be inferred from the analysis of linearized Boltzmann equation done in our other work [51]. There, we found three gapless modes (two sound modes denoted by $\sigma = \pm 1$ and the heat mode denoted by 0) with dispersion relations that at small $k$ behave as $\mathrm{Im}(\omega_{\sigma,0}(k)) = D_{\sigma,0}k^2$, where the diffusion constants are

$$D_0 = \frac{\kappa}{\varrho c_P}, \qquad D_\sigma = \frac{1}{2\varrho}\left[\left(\frac{1}{c_V} - \frac{1}{c_P}\right)\kappa + \zeta\right], \tag{155}$$

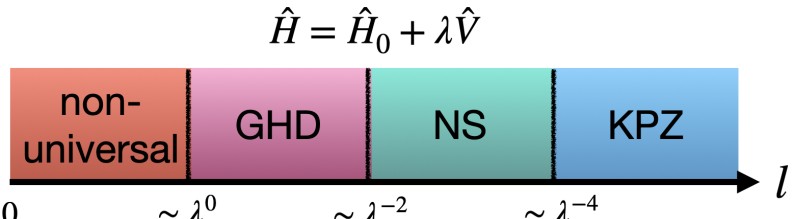

Figure 1: Hydrodynamic universality classes of the nearly integrable system across different length scales $l$. For $0 < l \lesssim l_0$, where $l_0 \sim \lambda^0$ is the microscopic length scale of integrable Hamiltonian $\hat{H}_0$, the hydrodynamics is not applicable and system behaves in non-universal way. At length scales $l_0 \ll l \ll l^*_{\text{NS/GHD}}$, where $l^*_{\text{NS/GHD}} \sim \lambda^{-2}$ the gas can be effectively described by GHD. For $l^*_{\text{NS/GHD}} \ll l \ll l^*_{\text{KPZ/NS}}$, where $l^*_{\text{KPZ/NS}} \sim \lambda^{-4}$ the gas is in NS hydrodynamic regime with transport coefficients computed in present work. Finally, at the largest length scales $l^*_{\text{KPZ/NS}} \ll l$ the system is characterized by KPZ anomalous transport, as expected for generic non-integrable one-dimensional fluids with momentum conservation.

where $c_P$ is specific heat at constant pressure. As mentioned earlier both viscosity and thermal conductivity have two contributions

$$\kappa = \kappa_{\mathfrak{D}} + \kappa_{\mathcal{I}}, \qquad \zeta = \zeta_{\mathfrak{D}} + \zeta_{\mathcal{I}}. \tag{156}$$

It is clear that transport coefficients stemming from $\mathfrak{D}$ do not depend on integrability-breaking parameter $\lambda$, as visible from (116). On the other hand, the contributions from $\mathcal{I}$ scale as $\lambda^{-2}$, as can be examined from (143) and (132). Therefore, for $\lambda$ small, contributions from $\mathcal{I}$ dominate over $\mathfrak{D}$ contributions leading to an overall scaling of diffusion constants

$$D_{0,\sigma} \sim \lambda^{-2}. \tag{157}$$

We emphasize here that this scaling will be crucial also for the analysis of crossover between NS diffusion and KPZ superdiffusion.

The NS description breaks down, when imaginary part of frequencies become comparable to the gap $\gamma$ of linearized Boltzmann kernel, proportional to $\lambda^2$ as the kernel itself. A comparison $\text{Im}(\omega_{\sigma,0}(k)) = D_{\sigma,0}k^2 \sim \gamma$ gives the same crossover length scale (154) as the simple argument with Knudsen number. At smaller length scales, the GHD theory applies [51]. Obviously, it inevitably has to break down to non-universal dynamics around $l \approx l_0$, where $l_0$ is microscopic length scale of integrable Hamiltonian $\hat{H}_0$.

Having established that NS equations occur for $l \gg l^*_{\text{NS/GHD}}$ we address the issue of KPZ anomalous transport emerging at the largest scales. Our point of interest will be the behavior of two-point functions of hydrodynamic fields close to the stationary, thermal state. The standard theory for such purpose in the context of one-dimensional non-integrable systems is the NLFH [71], to which we now turn.

## 5.1 Nonlinear fluctuating hydrodynamics for one-dimensional fluids

The framework of Ref. [71] is developed in the specific context of anharmonic chains. However, as noted in appendix of that work, the same theory can be equally well applied to one-dimensional fluids with conserved particle number, momentum and energy. The symmetries of those inter-mode couplings, which lead to long-lived perturbations are exactly the same [70]. We adapt the notation of [71] with only slight differences.

We start with NS equations written in the form (6) and with no external potential $U(x) = 0$. We consider small deviations from equilibrium as

$$q_0(x,t) \to q_0 + \delta q_0(x,t), \quad q_1(x,t) \to 0 + \delta q_1(x,t), \quad q_2(x,t) \to q_2 + \delta q_2(x,t), \quad (158)$$

and expand the NS currents

$$
\begin{aligned}
\mathcal{J}_0^{\text{NS}} &= q_1, \\
\mathcal{J}_1^{\text{NS}} &= \frac{q_1^2}{q_0} + P - \zeta \partial_x(q_1/q_0), \\
\mathcal{J}_2^{\text{NS}} &= \frac{q_1 q_2}{q_0} + \frac{q_1}{q_0} P - \zeta \frac{q_1}{q_0} \partial_x(q_1/q_0) - \kappa \partial_x T,
\end{aligned}
\qquad (159)
$$

keeping terms linear and quadratic in deviations $\delta q$ as well as linear in derivatives. Derivative $\partial_x T$ can be expressed with derivatives of charges via relations (3) and (5). The general structure of the resulting hydrodynamic balance equations is

$$\partial_t \delta q_\alpha + \partial_x \left( A_{\alpha\beta} \delta q_\beta + \frac{1}{2} H^\alpha_{\beta\gamma} \delta q_\beta \delta q_\gamma - \tilde{D}_{\alpha\beta} \partial_x \delta q_\beta + \tilde{B}_{\alpha\beta} \xi_\beta \right) = 0, \qquad (160)$$

with $A_{\alpha\beta} = \frac{\partial \mathcal{J}_\alpha^{\text{NS}}}{\partial q_\beta}$, $H^\alpha_{\beta\gamma} = \frac{\partial^2 \mathcal{J}_\alpha^{\text{NS}}}{\partial q_\beta \partial q_\gamma}$, and matrix $\tilde{D}$ involves terms with transport coefficients $\zeta, \kappa$. We have also added noise term $\tilde{B}$ representing fluctuations of the currents satisfying

$$\langle \xi_\alpha(x,t) \rangle = 0, \qquad \langle \xi_\alpha(x,t) \xi_\beta(x',t') \rangle = \delta_{\alpha\beta} \delta(x-x') \delta(t-t'). \qquad (161)$$

Moreover, we have

$$\langle \delta q_\alpha(x,t) \rangle = 0, \qquad \langle \delta q_\alpha(x,t) \delta q_\beta(x',t) \rangle = C_{\alpha\beta} \delta(x-x'), \qquad (162)$$

where $C_{\alpha\beta} \equiv \mathcal{C}_{\alpha\beta}$ is the susceptibility matrix, which fulfills $AC = CA^T$. The strength of the noise is fixed by fluctuation-dissipation relation $\tilde{D}C + C\tilde{D}^T = \tilde{B}\tilde{B}^T$. Equation (160) forms a starting point for the theory of NLFH. We will now recall main results of [71] for the correlation functions in that theory, which will allow us to understand the crossover between the NS and KPZ regimes.

We start with finding the normal modes (two sound modes denoted $\sigma = \pm 1$ and heat mode denoted by 0) by diagonalizing the matrix $A_{\alpha\beta}$ with transformation $R$ such that $RCR^T = 1$ and $RAR^{-1} = \text{diag}(-c, 0, c)$, where $c$ is sound velocity. Writing the equation in the normal mode basis $\vec{\phi} = R\delta\vec{q}$ we arrive at

$$\partial_t \phi_\alpha + \partial_x \left( c_\alpha \phi_\alpha + G^\alpha_{\beta\gamma} \phi_\beta \phi_\gamma - D_{\alpha\beta} \partial_x \phi_\beta + B_{\alpha\beta} \xi_\beta \right) = 0, \qquad (163)$$

with $c_\alpha \in \{-c, 0, c\}$ and where matrices $G, D, B$ follow from rotations of corresponding matrices $H, \tilde{D}, \tilde{B}$ with $R$. We are interested in two-point correlations of the fields $\phi_\alpha$

$$f_\alpha(x,t) = \langle \phi_\alpha(x,t) \phi_\alpha(0,0) \rangle, \qquad f_\alpha(x,0) = \delta(x), \qquad (164)$$

and analyze mode-coupling equations [71] giving approximate dynamics of $f_\alpha(x,t)$

$$\partial_t f_\alpha(x,t) = (-c_\alpha \partial_x + D_\alpha \partial_x^2) f_\alpha(x,t) + \int_0^t ds \int dy\, f_\alpha(x-y, t-s) \partial_y^2 M_{\alpha\alpha}(y,s), \qquad (165)$$

where $D_\alpha = D_{\alpha\alpha}$ and memory kernel reads

$$M_{\alpha\alpha}(x,t) = 2 \sum_{\beta,\gamma=0,\pm 1} (G^\alpha_{\beta\gamma})^2 f_\beta(x,t) f_\gamma(x,t). \qquad (166)$$

It is expected that the sound modes will move with velocity $c$ and therefore, after a sufficiently large time $f_\beta(x,t)f_\gamma(x,t) \approx 0$ for $\beta \neq \gamma$. Therefore, in this diagonal approximation

$$M_{\alpha\alpha}(x,t) \approx M_\alpha^{\mathrm{dg}} = 2\sum_{\gamma=0,\pm1}(G_{\gamma\gamma}^\alpha)^2 f_\gamma(x,t)^2. \tag{167}$$

In general, for one-dimensional fluids $G$ couplings have the following symmetries, which will be important for the structure of mode-coupling equations

$$G_{\beta\gamma}^\alpha = G_{\gamma\beta}^\alpha, \qquad G_{00}^0 = 0, \qquad G_{\sigma\sigma}^0 = -G_{-\sigma-\sigma}^0 \neq 0, \qquad G_{\alpha\beta}^\sigma = -G_{-\alpha-\beta}^{-\sigma} \neq 0. \tag{168}$$

We stress that $G$ couplings follow from thermodynamics given by TBA and thus, do not depend on integrability-breaking parameter $\lambda$, as predicted by ChE. With this, we move to the analysis of sound-sound correlation functions through mode-coupling equation.

## 5.2 Sound-sound correlation function

Due to the fact that peak travels with $c$, the only term kept in memory kernel corresponds to the same sound mode – for others, the overlap $f_\alpha \partial_y^2 M_{\alpha\alpha}$ is small. Hence, only one coupling is relevant and we have

$$\partial_t f_\sigma(x,t) = (-c_\sigma \partial_x + D_\sigma \partial_x^2)f_\sigma(x,t) + 2(G_{\sigma\sigma}^\sigma)^2 \int_0^t \mathrm{d}s \int \mathrm{d}y \, f_\sigma(x-y,t-s)\partial_y^2 f_\sigma(y,s)^2. \tag{169}$$

We go to Fourier space in position $f_\sigma(x,t) = \int \mathrm{d}k \, e^{ikx}\hat{f}_\sigma(k,t)$, remove the ballistic part of the evolution by replacing $\hat{f}_\sigma(k,t) \to e^{-ikc_\sigma t}\hat{f}_\sigma(k,t)$ and find

$$\partial_t \hat{f}_\sigma(k,t) = -k^2\left(D_\sigma \hat{f}_\sigma(k,t) + 2(G_{\sigma\sigma}^\sigma)^2 \int_0^t \mathrm{d}s \, \hat{f}_\sigma(k,t-s) \int \mathrm{d}q \, \hat{f}_\sigma(q,s)\hat{f}_\sigma(k-q,s)\right). \tag{170}$$

To make some progress with this equation we consider solutions in the scaling form

$$\hat{f}_\sigma(k,t) = p_\sigma(\Lambda_\sigma k^{\nu_\sigma}t), \tag{171}$$

where

- NS diffusion is $\nu_\sigma = 2$, $p_\sigma(w) = e^{-w}$ and $\Lambda_\sigma = D_\sigma \sim O(\lambda^{-2})$,

- KPZ superdiffusion is $\nu_\sigma = 3/2$, $p_\sigma(w) = \hat{f}_{\mathrm{mc}}(w)$ and $\Lambda_\sigma = 2\sqrt{2}G_{\sigma\sigma}^\sigma \sim O(1)$.

Here $\hat{f}_{\mathrm{mc}}$ is solution to fixed-point equation (178), which turns out to be close to KPZ scaling function [71]. Introducing new integration variables $\mu = \Lambda_\sigma k^{\nu_\sigma}s$ and then $r = \mu(q/k)^{\nu_\sigma}$ we arrive at the following expression (scaling variable is $w = \Lambda_\sigma k^{\nu_\sigma}t$)

$$
\begin{aligned}
k^{\nu_\sigma}\Lambda_\sigma p_\sigma'(w) = -k^2\Bigg[ &D_\sigma p_\sigma(w) + \frac{2(G_{\sigma\sigma}^\sigma)^2}{\nu_\sigma \Lambda_\sigma}k^{1-\nu_\sigma}\int_0^w \mathrm{d}\mu\, \mu^{-1/\nu_\sigma}p_\sigma(w-\mu) \\
&\times \int \mathrm{d}r\, r^{\frac{1-\nu_\sigma}{\nu_\sigma}}p_\sigma(r)p_\sigma\left(\mu\left(1-(r/\mu)^{1/\nu_\sigma}\right)^{\nu_\sigma}\right)\Bigg].
\end{aligned}
\tag{172}
$$

First, let us see what happens when the diffusive scaling function is assumed. We plug the corresponding values of $\nu_\sigma$ and $\Lambda_\sigma$ getting

$$
\begin{aligned}
k^2 D_\sigma p_\sigma'(w) = -k^2\Bigg[ &D_\sigma p_\sigma(w) + \frac{(G_{\sigma\sigma}^\sigma)^2}{D_\sigma}k^{-1}\int_0^w \mathrm{d}\mu\, \mu^{-1/2}p_\sigma(w-\mu) \\
&\times \int \mathrm{d}r\, r^{-\frac{1}{2}}p_\sigma(r)p_\sigma\left(\mu\left(1-(r/\mu)^{1/2}\right)^2\right)\Bigg].
\end{aligned}
\tag{173}
$$

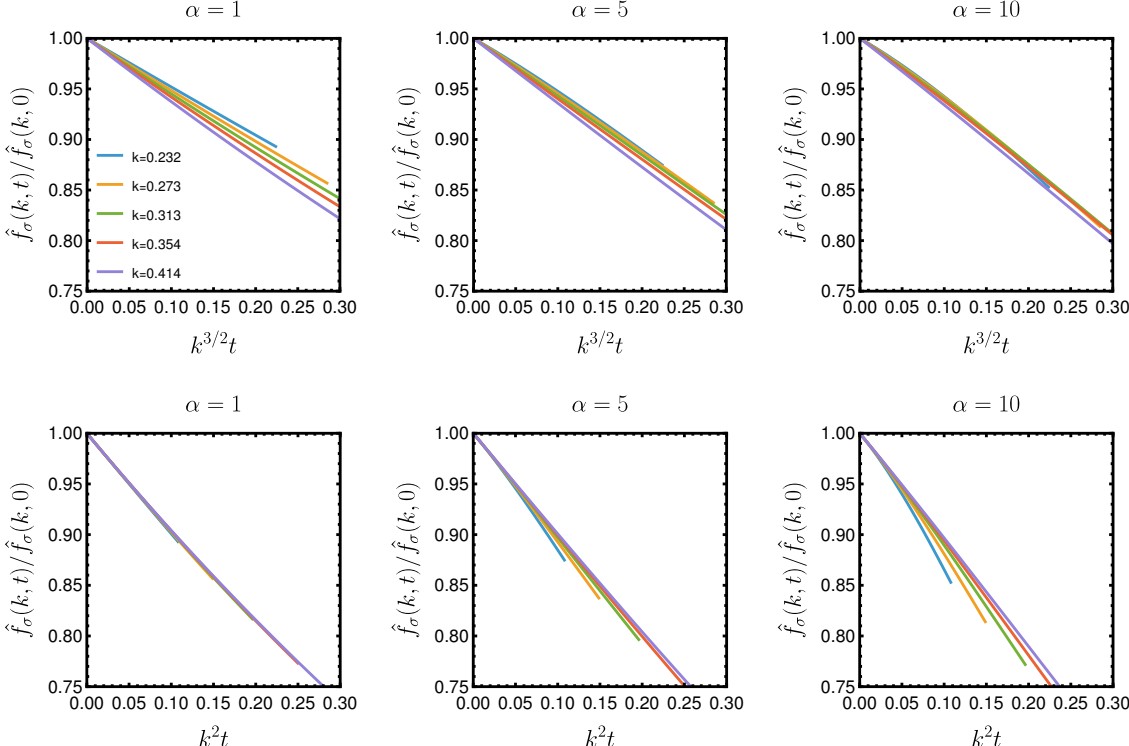

Figure 2: Dynamics of sound-sound correlation function $\hat{f}_\sigma(k,t)$. The initial state is a bimodal Gaussian distribution with variance defining characteristic inverse length scale of the state $k_c \approx 1/3$. In the equation (170), we rescaled $k$ by $k_c$ and time by $(k_c D_\sigma^2)^{-1}$ finding a single parameter $\alpha = 2(G_{\sigma\sigma}^\sigma)^2/(D_\sigma^2 k_c)$ controlling the relative strength of diffusive and mode-coupling terms in (170). Upper panel: correlator plotted versus anomalous scaling variable $k^{3/2}t$, lower panel: the same data against diffusive scaling variable $k^2 t$. For smaller $\alpha$, we recover diffusive behavior and good collapse to $k^2 t$ variable, whereas for large $\alpha$, where mode-coupling term dominates, we recover anomalous scaling as predicted by our analysis.

We see that due to large diffusion constant ($D_\sigma \sim \lambda^{-2}$), at finite, but small $k$ the first term on RHS may dominate the mode-coupling contribution (recall that $G$ coupling do not depend on $\lambda$). The characteristic value of $k$, for which this happens may be found by comparing the two terms. This corresponds to $D_\sigma \approx 1/(D_\sigma k)$ leading to an estimate

$$k \sim \lambda^4. \tag{174}$$

Above this characteristic value of $k$, diffusion dominates and mode interactions can be neglected. In this regime we recover standard equation for diffusion, here written in scaling variable $w$

$$p_\sigma'(w) = -p_\sigma(w). \tag{175}$$

Similar analysis may be performed assuming the superdiffusive form of the scaling function. Let us estimate when the purely diffusive behavior will break down to superdiffusion. We plug the corresponding values of $\nu_\sigma$ and $\Lambda_\sigma$ getting

$$k^{3/2}2\sqrt{2}G_{\sigma\sigma}^\sigma p_\sigma'(w) = -k^2\Bigg[D_\sigma p_\sigma(w)\frac{2G_{\sigma\sigma}^\sigma}{3\sqrt{2}}k^{-1/2}\int_0^w \mathrm{d}\mu\,\mu^{-2/3}p_\sigma(w-\mu)$$
$$\times \int \mathrm{d}r\,r^{-\frac{1}{3}}p_\sigma(r)p_\sigma\Big(\mu\big(1-(r/\mu)^{2/3}\big)^{3/2}\Big)\Bigg]. \tag{176}$$

Again we can estimate the crossover $k$, below which we can drop the first term with diffusion as

$$D_\sigma \approx k^{-1/2}, \tag{177}$$

leading again to $k \sim \lambda^4$ consistently with the previous analysis. Below these values of $k$, mode-coupling dominates. Neglecting now the diffusive term, we find the following equation

$$p'_\sigma(w) = -\frac{1}{6} \int_0^w \mathrm{d}\mu \mu^{-2/3} p_\sigma(w - \mu) \int \mathrm{d}r r^{-\frac{1}{3}} p_\sigma(r) p_\sigma \left( \mu \left( 1 - (r/\mu)^{2/3} \right)^{3/2} \right), \tag{178}$$

which gives a scaling function, which turns out to be close to KPZ scaling function [71].

In summary, we have found that above inverse length scales $k \sim \lambda^4$, the diffusion dominates over the mode-coupling term and sound-sound correlators are described by the standard diffusion equation with diffusion constant determined by transport coefficients found in this work (155). Therefore, we estimate the crossover length scale above which KPZ superdiffusion emerges as

$$l^*_{\mathrm{KPZ/NS}} \sim \lambda^{-4}. \tag{179}$$

In principle, there are additional terms in mode-coupling equation, but they have a subleading role. We present a short analysis in Appendix B.1.

The conclusions from analytical analysis of mode-coupling equations can be supported by numerical solutions. In Fig. 2 we present dynamics of sound-sound correlation function for different values of parameter $\alpha = 2(G^\sigma_{\sigma\sigma})^2/(D^2_\sigma k_c)$, where $k_c$ is characteristic inverse length-scale of initial state. For sufficiently small $\alpha$, we observe diffusive behavior, whereas for large $\alpha$ we find the collapse using scaling variable $k^{3/2}t$.

In a similar fashion, one can analyze equation for heat-heat correlations, we shortly discuss that in Appendix B.2. One finds then, the mode-coupling terms are relevant only at even larger scales $\sim \lambda^{-6}$. Thus, we keep length scale $\sim \lambda^{-4}$ as a scale for the crossover between the NS and KPZ regimes.

## 5.3 Additional remarks

Let us conclude these results with a few more comments. The NLFH predicts long-time tails in current-current correlation functions [86]. Thus, transport coefficients given by Green-Kubo relations are infinite as the formulas involve integration from $t = 0$ to $t = \infty$ probing all the scales in the system. However, the *bare transport coefficients* [87], which enter the equations of NLFH via diffusion matrix $D$ are finite and have to be finite in order to recover the KPZ superdiffusion [70]. Our work provides microscopic formulas for them as derived from ChE theory applied to the GHD-Boltzmann equation. Even though the formulas for correlators in KPZ regime do not depend on them, they are crucial for the NS dynamics in intermediate length scales (which in practice may be very large). Interestingly, as the KPZ correlation functions depend only on thermodynamic quantities fixed by TBA, namely $G$ couplings and sound velocity, at the largest scale the dynamics of the system does not depend on the details of integrability-breaking perturbation.

It is also worth mentioning that Refs. [83–85] studying 1D nearly integrable systems found a normal transport across a large window of system sizes. This is in agreement with conclusions of our work, which is based on NLFH. The general picture of nearly-integrable dynamics are quasiparticles with parametrically (in integrability-breaking parameter $\lambda$) long relaxation times. From kinetic theory point of view this naturally leads to large bare transport coefficients, proportional to thermalization time. We emphasize that their large values are the only ingredients needed to arrive at our conclusions using NLFH. Thus, these observations point towards a more general implications of our analysis.

Lastly, we note that our framework incorporates external trapping, which breaks the momentum conservation. In such scenario, the transport is expected to be diffusive with finite thermal conductivity [88–91].

# 6 Conclusions and discussion

In this work, we have addressed a problem of non-equilibrium dynamics of nearly integrable quantum gases starting with the description of the system at the level of the generalized hydrodynamics. We have shown that, in the presence of integrability-breaking collision integral the problem can be studied by adopting the methods of kinetic theory. This allowed us to derive the NS equations, valid at large space-time-scales, together with the exact expressions for the transport coefficients.

The computations presented here complement findings of [51] where the NS equations were derived for a linearized theory and applied to a concrete setup of coupled Lieb-Liniger gases. Here instead, the full non-linear problem was analyzed by generalizing the ChE method known from the kinetic theory of dilute gases. Interestingly, we find that the transport coefficients in both approaches, linearized and ChE, yield the same expressions for the transport coefficients. We stress that derivation of the full, nonlinear hydrodynamics is of key importance to the study of crossover between NS diffusion and KPZ superdiffusion in our system. The theory of NLFH, on which our analysis is based, requires nonlinear contributions to the currents, which were absent in derivation presented in [51]. In addition to that, the ChE framework allowed us to incorporate the effects of external potential into the NS equations.

The resulting transport coefficients have two contributions reflecting the two types of interactions present in the system. There is an integrable interaction captured by the diffusion term of the unperturbed generalized hydrodynamics, and there is an integrability-breaking interaction described by the collision integral. Importantly, both contributions are non-perturbative in the strength of the integrable interactions and are valid beyond the dilute limit. As already pointed out in [51], the viscosity is generally non-zero. This should be contrasted with the kinetic theory of dilute gases which predicts always zero viscosity for 1d fluids.

One interesting direction in which the result of present work could be extended is the following. We could envision supplementing the NS equations by a dynamic equation for the slowest decaying mode [82]. This would extend the resulting description towards shorter timescales when more information on the integrable structure is relevant for the dynamics.

In the low temperature regime, the conventional hydrodynamics may emerge from GHD without integrability-breaking. In particular, the Euler-scale GHD becomes equivalent to conventional Euler hydrodynamics at zero temperature [79]. When the diffusive effects are incorporated into the GHD, the effective low-temperature description at early times is the NS hydrodynamics [92]. One then finds transport coefficients given by the contributions from integrable interactions $\kappa_{\mathfrak{D}}, \zeta_{\mathfrak{D}}$ solely.

Conventional hydrodynamics in one-dimensional momentum-conserving systems is known to be unstable at the largest space-time scales due to mode interactions [68–71]. The transport universality class changes then from diffusive to anomalous, KPZ class. The relevant theory of anomalous transport in such systems in the NLFH [71], which starts with 1D NS equations supplemented with noise terms. At the largest space-time scales, KPZ behavior of two-point functions is recovered.

We addressed this issue in Sec. 5, where we have shown the mode interactions are relevant only at the largest length scales $\sim \lambda^{-4}$, where $\lambda$ is the integrability-breaking parameter. At smaller scales, the leading effects are captured by NS hydrodynamics with finite transport coefficients determined by us in this work. More precisely, we have found that nearly inte-

grable quantum gases are characterized by three different hydrodynamic regimes: GHD at the smallest (but still macroscopic) length scales, NS in the intermediate regime, and KPZ at the largest scales. The superdiffusive correlation functions in the KPZ regime are determined by the couplings fixed with the thermodynamics given by TBA and do not depend on the details of integrability-breaking perturbation.

A possible direction for future work is the comparison of the results from GHD-Boltzmann equation against NS hydrodynamics with transport coefficients computed in this work. As the numerical solution of GHD-Boltzmann equation suffers from costly computation of Boltzmann integrals, it seems natural to consider the so-called relaxation time approximation to collision integral. In Ref. [52] such equation was solved showing signatures of diffusive spreading characteristic to the NS dynamics. We also note that NS equations were proven useful in description of dynamics of 1d gases as recently demonstrated in Refs. [93, 94] comparing numerical simulations of blast waves initial conditions with NS hydrodynamics.

In this work, we have focused on a theory with Galilean invariance which determines the structure of the resulting hydrodynamic equations to be of the NS form. However, the developed methods are applicable to other systems described by the generalized hydrodynamics. The most natural and important are integrable spin chains whose non-equilibrium dynamics can be efficiently studied with tensor networks methods.

# Acknowledgments

We thank Jacopo De Nardis, Alvise Bastianello, Robert Konik, Leonardo Biagetti, Piotr Szymczak for insightful discussions on this and closely related topics.

**Funding information**  MŁ and MP acknowledge support by the National Science Centre (NCN), Poland via projects 2018/31/D/ST3/03588 and 2022/47/B/ST2/03334.

# A  Chapman-Enskog theory: Additional formulas and results

## A.1  Thermodynamics with hydrodynamic matrices

Starting from GHD-Boltzmann equation we naturally encounter matrix elements of $\mathcal{C}, \mathcal{B}$ matrices evaluated in thermal states. To recast the equations which follow from ChE expansion into universal NS form it is useful to connect these quantities to thermodynamics of the system. The calculations are presented in Supplemental Material of [51]. Here, we list the most relevant expressions which are formulas for adiabatic compressibility $\varkappa_S$, isothermal compressibility $\varkappa_T$, sound velocity $c$ and specific heats at constant volume $c_V$ and pressure $c_P$:

$$\varkappa_S = \frac{1}{\varrho}\left(\frac{\partial \varrho}{\partial P}\right)_s = T\frac{\bar{\mathcal{C}}_{2,2}\bar{\mathcal{C}}_{0,0} - \bar{\mathcal{C}}_{2,0}^2}{\bar{\mathcal{B}}_{1,0}^2\bar{\mathcal{C}}_{2,2} + \bar{\mathcal{B}}_{2,1}^2\bar{\mathcal{C}}_{0,0} - 2\bar{\mathcal{B}}_{1,0}\bar{\mathcal{B}}_{2,1}\bar{\mathcal{C}}_{2,0}}, \tag{A.1}$$

$$\varkappa_T = \frac{1}{\varrho}\left(\frac{\partial \varrho}{\partial P}\right)_T = \frac{1}{\varrho}\frac{\bar{\mathcal{C}}_{0,0}}{\bar{\mathcal{B}}_{1,0}}, \tag{A.2}$$

$$c = \sqrt{\frac{1}{\mathcal{B}_{1,0}}\frac{\mathcal{B}_{1,0}^2\mathcal{C}_{2,2} + \mathcal{B}_{2,1}^2\mathcal{C}_{0,0} - 2\mathcal{B}_{1,0}\mathcal{B}_{2,1}\mathcal{C}_{2,0}}{\mathcal{C}_{2,2}\mathcal{C}_{0,0} - \mathcal{C}_{2,0}^2}}, \tag{A.3}$$

$$c_V = \frac{1}{\varrho}\left(\frac{\partial e}{\partial T}\right)_\varrho = \frac{1}{T}\frac{\bar{\mathcal{C}}_{2,2}\bar{\mathcal{C}}_{0,0} - \bar{\mathcal{C}}_{2,0}^2}{\bar{\mathcal{C}}_{0,0}\bar{\mathcal{B}}_{1,0}}, \tag{A.4}$$

$$c_P = \left(\frac{\partial s}{\partial T}\right)_P = \frac{1}{T}\frac{\bar{\mathcal{B}}_{1,0}^2\bar{\mathcal{C}}_{2,2} + \bar{\mathcal{B}}_{2,1}^2\bar{\mathcal{C}}_{0,0} - 2\bar{\mathcal{B}}_{1,0}\bar{\mathcal{B}}_{2,1}\bar{\mathcal{C}}_{2,0}}{\bar{\mathcal{B}}_{1,0}^3}, \tag{A.5}$$

$$\sqrt{c_P/c_V - 1} = \frac{\bar{\mathcal{B}}_{2,1}\bar{\mathcal{C}}_{0,0} - \bar{\mathcal{B}}_{1,0}\bar{\mathcal{C}}_{2,0}}{\bar{\mathcal{B}}_{1,0}\sqrt{\bar{\mathcal{C}}_{2,2}\bar{\mathcal{C}}_{0,0} - \bar{\mathcal{C}}_{2,0}^2}}, \qquad \bar{\mathcal{B}}_{1,0} = \varrho T. \tag{A.6}$$

## A.2 Galilean invariance of the Chapman-Enskog equations

In this Appendix, we demonstrate that the ChE equations are invariant under the Galilean boost. The equations, including the constitutive relations, are

$$\Gamma w_c = \mathcal{B}\hbar_c - \sum_{b=0}^{2}\mathcal{C}\hbar_b\mathcal{A}_{bc}^{\text{Eul}}, \qquad \hbar_a\mathcal{C}w_c = 0, \qquad a,c = 0,1,2. \tag{A.7}$$

To prove the invariance we start with the transformation rules of the $(3 \times 3)$-dimensional hydrodynamic matrices at the Euler scale,

$$\mathcal{C}^{\text{Eul}} = g(u)\bar{\mathcal{C}}^{\text{Eul}}g^T(u), \qquad \mathcal{B}^{\text{Eul}} = g(u)\bar{\mathcal{B}}^{\text{Eul}}g^T(u) + ug(u)\bar{\mathcal{C}}^{\text{Eul}}g^T(u), \tag{A.8}$$

from which follows

$$\left(\mathcal{C}^{\text{Eul}}\right)^{-1} = g^T(-u)\left(\bar{\mathcal{C}}^{\text{Eul}}\right)^{-1}g(-u), \tag{A.9}$$

and

$$\mathcal{A}^{\text{Eul}} = g^T(-u)\bar{\mathcal{A}}^{\text{Eul}}g^T(u) + u. \tag{A.10}$$

This allows us to rewrite the right hand side as

$$\mathcal{B}\hbar_c - \sum_{b=0}^{2}\mathcal{C}\hbar_b\mathcal{A}_{bc}^{\text{Eul}} = b_u^{-1}\left(\bar{\mathcal{B}}b_u\hbar_c - \sum_{b=0}^{2}\bar{\mathcal{C}}b_u\hbar_b\left(g^T(-u)\bar{\mathcal{A}}^{\text{Eul}}g^T(u)\right)_{bc}\right). \tag{A.11}$$

We can now use that $b_u\sum_b\hbar_b(g^T(-u))_{bc} = \hbar_c$ and $b_u\hbar_c = \sum_b g_{cb}(u)\hbar_b$ to obtain

$$\mathcal{B}\hbar_c - \sum_{b=0}^{2}\mathcal{C}\hbar_b\mathcal{A}_{bc}^{\text{Eul}} = b_u^{-1}\sum_{b=0}^{2}\left(\bar{\mathcal{B}}\hbar_b - \sum_{d=0}^{2}\bar{\mathcal{C}}\hbar_d(\bar{\mathcal{A}}^{\text{Eul}})_{db}\right)\left(g^T(u)\right)_{bc}. \tag{A.12}$$

Defining $\Gamma = b_u^{-1}\bar{\Gamma}b_u$ and $w_c = b_u^{-1}\bar{w}_d(g^T(u))_{dc}$ we find

$$\bar{\Gamma}\bar{w}_c = \bar{\mathcal{B}}\hbar_c - \sum_{b=0}^{2}\bar{\mathcal{C}}\hbar_b\bar{\mathcal{A}}_{bc}^{\text{Eul}}, \qquad \hbar_a\bar{\mathcal{C}}\bar{w}_c = 0, \qquad a,c = 0,1,2, \tag{A.13}$$

as reported in (130).

## A.3 Equations for transport coefficients in the orthonormal basis

In this Appendix, we show explicitly that integral equations (132) are the same as equations presented in our earlier work [51], where a basis orthonormal with respect to the hydrodynamic inner product with matrix $\bar{\mathcal{C}}$, $(h_n|h_m) = \delta_{nm}$ was used.

Transport coefficients $\zeta_{\mathcal{I}}, \kappa_{\mathcal{I}}$ are determined from formulas (143) with functions $\bar{w}_1, \bar{w}_2$ determined from integral equations (132). The orthonormal basis $\{h_n\}_{n\in\mathbb{N}}$ is constructed using Gram-Schmidt procedure on the ultralocal basis $\{\hbar_n\}_{n\in\mathbb{N}}$. The first three elements are

$$h_0 = \frac{1}{\sqrt{\bar{\mathcal{C}}_{0,0}}}\hbar_0, \qquad h_1 = \frac{1}{\sqrt{\bar{\mathcal{C}}_{1,1}}}\hbar_1, \qquad h_2 = \sqrt{\frac{\bar{\mathcal{C}}_{0,0}}{\bar{\mathcal{C}}_{2,2}\bar{\mathcal{C}}_{0,0} - \bar{\mathcal{C}}_{2,0}^2}}\left(\hbar_2 - \frac{\bar{\mathcal{C}}_{2,0}}{\bar{\mathcal{C}}_{0,0}}\hbar_0\right). \tag{A.14}$$

Rewriting equations (132) and using thermodynamic expression from Appendix A.1 we find

$$\Gamma \bar{w}_1 = \sqrt{\varrho T}\bar{\mathcal{B}}h_1 - \sqrt{T/\varkappa_T}\bar{\mathcal{C}}(h_0 + \sqrt{c_P/c_V - 1}\,h_2)\,, \tag{A.15}$$

$$\Gamma \bar{w}_2 = T\sqrt{c_v\varrho}\,\bar{\mathcal{B}}(h_2 - \sqrt{c_P/c_V - 1}\,h_0)\,, \tag{A.16}$$

defining now

$$\phi_\zeta = \bar{w}_1/T\,, \qquad \phi_\kappa = \bar{w}_2/T^2\,, \tag{A.17}$$

we recover the equations given in the main text of [51]

$$\bar{\Gamma}\phi_\zeta = \sqrt{\beta\varrho}\bar{\mathcal{B}}h_1 - \sqrt{\beta/\varkappa_T}\bar{\mathcal{C}}(h_0 + \sqrt{c_P/c_V - 1}\,h_2)\,, \tag{A.18}$$

$$\bar{\Gamma}\phi_\kappa = \beta\sqrt{\varrho c_V}\bar{\mathcal{B}}(h_2 - \sqrt{c_P/c_V - 1}\,h_0)\,, \tag{A.19}$$

and with the same transport coefficients. Indeed, expressions (143) become then

$$\zeta_{\mathcal{I}} = \hbar_1\bar{\mathcal{B}}\phi_\zeta\,, \qquad \kappa_{\mathcal{I}} = \hbar_2\bar{\mathcal{B}}\phi_\kappa\,, \tag{A.20}$$

as in [51].

# B  Mode-coupling equations: Additional results

## B.1  Analysis of the subleading mode-coupling terms

In this part we investigate subleading corrections to sound-sound mode-coupling equation which stem from additional interactions. Before, we were considering memory kernel in the form

$$M_1(x,t) = 2(G^1_{11})^2 f_1(x,t)f_1(x,t)\,, \tag{B.1}$$

now, we include more terms, i.e.

$$M_1(x,t) = 2\Big[(G^1_{11})^2 f_1(x,t)f_1(x,t) + (G^1_{00})^2 f_0(x,t)f_0(x,t) + (G^1_{-1-1})^2 f_{-1}(x,t)f_{-1}(x,t)\Big]. \tag{B.2}$$

Let us look on the mode-coupling equation in Fourier space

$$\begin{aligned}
\partial_t \hat{f}_1(k,t) = &(-ikc - k^2 D_1)\hat{f}_1(k,t) \\
&- 2k^2 \int_0^t ds\,\hat{f}_1(k,t-s)\Big[(G^1_{11})^2 \int dq\,\hat{f}_1(q,s)\hat{f}_1(k-q,s) \\
&+ (G^1_{00})^2 \int dq\,\hat{f}_0(q,s)\hat{f}_0(k-q,s) + (G^1_{-1-1})^2 \int dq\,\hat{f}_{-1}(q,s)\hat{f}_{-1}(k-q,s)\Big].
\end{aligned} \tag{B.3}$$

We write the phases explicitly replacing

$$\hat{f}_1(k,t) \to e^{-ikct}\hat{f}_1(k,t)\,, \qquad \hat{f}_{-1}(k,t) \to e^{ikct}\hat{f}_1(k,t)\,, \tag{B.4}$$

and get

$$\begin{aligned}
\partial_t \hat{f}_1(k,t) = -k^2\Big[&D_1\hat{f}_1(k,t) + 2\int_0^t ds\,\hat{f}_1(k,t-s) \\
&\times\Big((G^1_{11})^2 \int dq\,\hat{f}_1(q,s)\hat{f}_1(k-q,s) + (G^1_{00})^2 e^{ikcs} \\
&\times \int dq\,\hat{f}_0(q,s)\hat{f}_0(k-q,s) + (G^1_{-1-1})^2 e^{2ikcs}\int dq\,\hat{f}_1(q,s)\hat{f}_1(k-q,s)\Big)\Big].
\end{aligned} \tag{B.5}$$

The idea here is to assume diffusive scaling of heat-heat and sound-sound correlators and estimate, under this assumption, the role of additional interactions. Thus we write

$$\hat{f}_0(k,t) = g_0\big((D_0 t)^{1/2}k\big), \qquad \hat{f}_1(k,t) = p_1(D_1 k^2 t) = g_1\big((D_1 t)^{1/2}k\big), \qquad \text{(B.6)}$$

where, for convenience we introduced two representations of the same function through $p_1(w) = e^{-w}$ and $g_{0,1}(w) = e^{-w^2}$. In RHS of (B.5) we have three mode coupling terms and we already analyzed the first one in Sec. 5.2. For the second one we introduce $w = D_1 k^2 t$ change variables as $\bar{s} = ks$ and then $r = (D_0 \bar{s})^{1/2} k^{-1/2} q$ getting

$$\frac{2(G_{00}^1)^2}{D_0^{1/2} k^{1/2}} \int_0^{\frac{w}{D_1 k}} d\bar{s} \frac{e^{ic\bar{s}}}{\sqrt{\bar{s}}} p_1(w - D_1 k\bar{s}) \int dr \, g_0(r) g_0((D_0 \bar{s})^{1/2} k^{1/2} - r). \qquad \text{(B.7)}$$

We note now that due to our assumption $k \ll \lambda^2$ we also have $D_{0,1} k \ll 1$. This allows to extend the integration to $\infty$ and drop $k$-dependent terms in arguments of $p_1$ and $g_0$ getting

$$\frac{2(G_{00}^1)^2}{D_0^{1/2} k^{1/2}} \int_0^\infty d\bar{s} \frac{e^{ic\bar{s}}}{\sqrt{\bar{s}}} \int dr \, g_0(r) g_0(-r) p_1(w) := \frac{\mathfrak{C}_{00}^1}{D_0^{1/2} k^{1/2}} p_1(w), \qquad \text{(B.8)}$$

where $\mathfrak{C}_{00}^1 = \text{const}$. Similar analysis may be performed for the third term, one finds

$$\frac{2(G_{-1-1}^1)^2}{D_1^{1/2} k^{1/2}} \int_0^\infty d\bar{s} \frac{e^{2ic\bar{s}}}{\sqrt{\bar{s}}} \int dr \, g_1(r) g_1(-r) p_1(w) := \frac{\mathfrak{C}_{-1-1}^1}{D_1^{1/2} k^{1/2}} p_1(w), \qquad \text{(B.9)}$$

thus we can write the full mode-coupling equation as

$$k^2 D_1 p_1'(w) = -k^2 D_1 \Bigg[ p_1(w) + \frac{(G_{11}^1)^2}{D_1^2 k} \int_0^w d\mu \, \mu^{-1/2} p_1(w - \mu) \int dr \, r^{-\frac{1}{2}} p_1(r) p_1\Big(\mu \big(1 - (r/\mu)^{1/2}\big)^2\Big) \\ + \frac{\mathfrak{C}_{00}^1}{D_1 D_0^{1/2} k^{1/2}} p_1(w) + \frac{\mathfrak{C}_{-1-1}^1}{D_1^{3/2} k^{1/2}} p_1(w) \Bigg].$$
$$\text{(B.10)}$$

We clearly see, that for inverse length scales much bigger than $\sim \lambda^4$ (which means $D_{0,1}^2 k \gg 1$) both the second term and also the two additional ones are negligible comparing to the first one, which gives standard diffusion. Actually, the two additional terms become comparable to diffusion at inverse lengths scales $\sim \lambda^6$ and thus are subleading corrections.

## B.2 Heat-heat correlation function

The heat-heat correlator is characterized by a different exponent and is described by Levy-$\frac{5}{3}$ distribution, instead of KPZ scaling function. This is essentially due to the fact that for heat mode self-coupling is absent as $G_{00}^0 = 0$ [71] and one has to look for subleading terms in the memory kernel. Including the coupling to the sound-modes gives

$$\partial_t f_0(x,t) = D_0 \partial_x^2 f_0(x,t) + 2 \sum_\sigma (G_{\sigma\sigma}^0)^2 \int_0^t ds \int dy \, f_0(x-y, t-s) \partial_y^2 f_\sigma(y,s)^2. \qquad \text{(B.11)}$$

We start with the relevant equation in Fourier space. It reads (we write phase factors $e^{\pm ikcs}$ explicitly)

$$\partial_t \hat{f}_0(k,t) = -k^2 \Bigg( D_0 \hat{f}_0(k,t) + 4(G_{\sigma\sigma}^0)^2 \int_0^t ds \cos(kcs) \hat{f}_0(k, t-s) \int dq \, \hat{f}_\sigma(q,s) \hat{f}_\sigma(k-q,s) \Bigg). \qquad \text{(B.12)}$$

Once again we try with solution in scaling form

$$\hat{f}_0(k, t) = p_0(\Lambda_0 k^{\nu_0} t), \tag{B.13}$$

where

- NS diffusion is $\nu_0 = 2$, $p_0(w) = e^{-w}$ and $\Lambda_0 = D_0 \sim O(\lambda^{-2})$,

- Levy-$\frac{5}{3}$ superdiffusion is $\nu_0 = 5/3$, $p_0(w) = e^{-|w|}$ and $\Lambda_\sigma = \Lambda_0^{\text{Levy}} \sim O(1)$.

Following similar calculation to the one presented for the case of sound-sound correlator and in [71] we find that there again is a crossover between normal diffusion and superdiffusion. This time however, we find that it happens at even larger scales

$$l_{\text{Levy/NS}}^{\text{heat}} \sim \lambda^{-6}. \tag{B.14}$$

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
