# Peer review of "Chapman-Enskog theory and crossover between diffusion and superdiffusion for nearly integrable quantum gases"

_SciPost Physics, doi:SciPost Phys. 18, 186 (2025)_

## Round 1 · Referee Report · Anonymous (Referee 1) · 2024-12-30

Strengths

  1. The problem considered in the paper is timely, given current research in the field.
  2. The theoretical background is outlined clearly, and the computations are explained in detail.
  3. The authors present a partial discussion of the potential limitations of the approach.

Weaknesses

  1. The authors give no explicit example of applying the formalism developed in the paper, which also means that the possible complications and issues arising in an application are not explored.
  2. Connected to the previous point, application to at least one system would be necessary to demonstrate the approach's range of validity.

Report

It is hard to assess whether the acceptance criteria are met due to the weaknesses of the paper. The derivation presented in the paper is a more or less straightforward application of Chapman-Enskog theory in the GHD framework. Whether this is a significant step forward depends on its applicability, which is hard to gauge given that the authors present no application of the formalism to any concrete system, and there is no comparison to some other approach, such as, e.g., a numerical simulation. This prevents assessing whether the journal expectations claimed by the authors are satisfied.

Additionally, without a concrete example, the potential complications of applying the formalism are left unclear.

Due to these weaknesses, I suggest transferring the paper to Scipost Physics Core, where it meets the application criteria and can be published without significant revision.

Requested changes

See the report.

Recommendation

Accept in alternative Journal (see Report)

  • validity: high
  • significance: good
  • originality: good
  • clarity: high
  • formatting: excellent
  • grammar: excellent

Author:  Maciej Łebek  on 2025-04-10  [id 5357]

(in reply to Report 1 on 2024-12-30)

Dear Referee,

Thank you for feedback and comments on our manuscript. In the following, we would like to address the specific points raised in the report.

1. The authors give no explicit example of applying the formalism developed in the paper, which also means that the possible complications and issues arising in an application are not explored. 2. Connected to the previous point, application to at least one system would be necessary to demonstrate the approach's range of validity

An application of the formalism developed here has been shown in the short letter referenced in the text in [51]. We consider there an experimentally relevant case of integrability breaking by long-range coupling between two Lieb-Liniger gases. This is an important setup for experiments with ultra-cold dipolar atomic gases. Thank you for that comment, we discuss the application of the formalism in the summary of our new version.

The derivation presented in the paper is a more or less straightforward application of Chapman-Enskog theory in the GHD framework.

We believe that presented result constitute an important generalization of the standard Chapman-Enskog procedure. There are two aspects: first, the thermodynamics is of a strongly correlated system, yet, as observed by the referee, the results still follow from more or less the same procedure. However, this is not guaranteed and in fact shows that the Chapmann-Enskog method does not rely on thermodynamics of a dilute gas as could be judged from standard applications. Second point, which is even more important, relates to the form of kinetic equations that are the starting point of the method. The GHD-Boltzmann equation, unlike the original Boltzmann equation, contains effective velocity, which is nonlinear functional of the state and makes the equation much more complicated. Moreover, it has a diffusive term. This diffusive term turns out to be effectively invariant under the Chapman-Enskog method (it remains of the same form). This shows, that there is no renormalization of the GHD diffusion constant due to integrating out the modes corresponding to higher conserved charges broken by the collision integral. This is important physical result showing that the transport coefficients have two separate contributions that can be assigned to two types of scatterings present in the system. Note that this additive character of the transport coefficient is hard to expect by looking at the GHD-Boltzmann equation. Finally, we would like to point out, that the present work represents a fairly new direction of research in employing methods of kinetic theory into a study of weakly perturbed integrable systems. A similar line reasoning lead us recently to generalize the famous BBGKY hierarchy to studies of such systems in [64]. Such a framework can be understood as an alternative way (as opposed to Fermi's Golden Rule) for deriving collision integrals, on which theory of the present article is based.

Due to these weaknesses, I suggest transferring the paper to Scipost Physics Core, where it meets the application criteria and can be published without significant revision.

We strongly believe that our article fulfills the acceptance criteria of SciPost Physics. Firstly, as discussed above, the application of the Chapman-Enskog method to the GHD-Boltzman equation solves an important, difficult and novel problem. Secondly, we would like to draw the attention of the Referee to the new section (Sec.5 in the present version), which presents new results about the crossover between Navier-Stokes diffusive transport and Kardar-Parisi-Zhang superdiffusion. The new section lays out foundations for the intuitions presented in summary section of the first version. It was motivated by the remarks of the Second Referee about the emergence of anomalous transport, a generic feature of non-integrable fluids in one-dimension. The section material presents new results, which are important for establishing a regime of weak integrability breaking. In particular, we give estimates for crossover length scales for different hydrodynamic universality classes displayed by the system. To indicate the importance of these results, we decided to change the title and abstract of our article. With this new material, not only we are able to fully address the criticism of the Second Referee, but we also added a significant value to the manuscript. We believe that with these changes, the paper is well-suited for publication in SciPost Physics.

Sincerely, Authors

---

## Round 1 · Referee Report · Anonymous (Referee 2) · 2024-12-31

Report

The authors develop a Chapman-Enskog theory for perturbed integrable systems. The Chapman-Enskog expansion is a textbook method for deriving the Navier-Stokes equations of fluid dynamics from the Boltzmann equation. However, the Chapman-Enskog theory is not a rigorous or controlled theory of hydrodynamics, does not yield exact results in practice and to my mind offers mostly qualitative insight into how perturbing a free (and thus trivially integrable) system gives rise to irreversible, diffusive behaviour.

In fact I believe that the authors’ results are invalid, because the broadening of sound modes in one dimension momentum-conserving systems is not diffusive, as the authors’ analysis would predict, but is instead well-known to exhibit z=3/2 KPZ-type scaling, as discussed e.g. in Ref. 79 and references therein. The heat mode would presumably also exhibit anomalous broadening. Thus I expect that all the dissipative transport coefficients obtained by the authors are in fact infinite! Only when the integrability-breaking perturbation is exactly zero, and there is no nonlinear mode coupling (which turns out to be guaranteed by the linear degeneracy property of GHD) is it possible to escape anomalous broadening.

I do not believe that the authors’ remarks in the conclusion adequately address this point, which is specific to the one-dimensional setting in which interacting integrable systems are known to occur. (It does not occur in three dimensions, which is why Chapman-Enskog theory is successful there.) Since the authors make no attempt to test the quantitative accuracy of their predictions, e.g. by numerically simulating a specific (presumably classical) model, I am skeptical that they are applicable in any spacetime regime, as the transport coefficients would all have to diverge as a function of time to be consistent with nonlinear fluctuating hydrodynamics.

Given this, I am afraid that I cannot recommend the current version of this paper for publication.

Recommendation

Reject

  • validity: -
  • significance: -
  • originality: -
  • clarity: -
  • formatting: -
  • grammar: -

Author:  Maciej Łebek  on 2025-04-10  [id 5358]

(in reply to Report 2 on 2024-12-31)

Dear Referee,

we are thankful for the feedback and comments on our manuscript, which motivated us to look for more quantitative understanding of the crossover between Navier-Stokes diffusion and Kardar-Parisi-Zhang superdiffusion, which before was just discussed in the summary of the previous version. In the following, we would like to address the specific raised points in your report.

The Referee writes:

The authors develop a Chapman-Enskog theory for perturbed integrable systems. The Chapman-Enskog expansion is a textbook method for deriving the Navier-Stokes equations of fluid dynamics from the Boltzmann equation. However, the Chapman-Enskog theory is not a rigorous or controlled theory of hydrodynamics, does not yield exact results in practice and to my mind offers mostly qualitative insight into how perturbing a free (and thus trivially integrable) system gives rise to irreversible, diffusive behaviour.

Our response:

We do not agree that Chapman-Enskog method gives only qualitative insight into behavior of conventional fluids. The method gives concrete predictions for viscosities and thermal conductivity of three-dimensional dilute gases. For example, as summarized in Chapter 3, Section 3.7 in [1], these predictions compare excellently with experimental data for many different gases across wide range of state parameters. For these reasons, we find the statement of the Referee unjustified.
What is more, it is also not true that Chapman-Enskog theory was applied only for weakly perturbed free systems. A textbook example of such application is the treatment of the Enskog theory of hard sphere gas, see for example [2].

The Referee writes:

In fact I believe that the authors’ results are invalid, because the broadening of sound modes in one dimension momentum-conserving systems is not diffusive, as the authors’ analysis would predict, but is instead well-known to exhibit z=3/2 KPZ-type scaling, as discussed e.g. in Ref. 79 and references therein. The heat mode would presumably also exhibit anomalous broadening. Thus I expect that all the dissipative transport coefficients obtained by the authors are in fact infinite! Only when the integrability-breaking perturbation is exactly zero, and there is no nonlinear mode coupling (which turns out to be guaranteed by the linear degeneracy property of GHD) is it possible to escape anomalous broadening.
I do not believe that the authors’ remarks in the conclusion adequately address this point, which is specific to the one-dimensional setting in which interacting integrable systems are known to occur. (It does not occur in three dimensions, which is why Chapman-Enskog theory is successful there.) Since the authors make no attempt to test the quantitative accuracy of their predictions, e.g. by numerically simulating a specific (presumably classical) model, I am skeptical that they are applicable in any spacetime regime, as the transport coefficients would all have to diverge as a function of time to be consistent with nonlinear fluctuating hydrodynamics.

Our response:

As noted by the Referee, we discussed the anomalous hydrodynamics in the conclusions. There we provided a number of references in which it is shown that standard transport coefficients of the Navier-Stokes equations exist in various context for the hydrodynamics of one dimensional fluids. The most straightforward result is for systems with broken momentum conservation, for example, by a confining potential. This is a relevant case for the Lieb-Liniger model and its variants as they serve an important role in modeling experiments with cold atomic gases where confining potentials are always present. However, even in homogeneous systems with the momentum conservation, the normal hydrodynamics might prevail over the anomalous if system sizes are small compared to the KPZ length scale. This has been observed for example in [82,84]. This point was further strengthened in [83,85] where importance of being close to an integrable point was highlighted, which is exactly our setup. Given that, we are convinced that our results for transport coefficients carry physical significance and disagree with the overall assessment of the validity of the approach presented in this work.

However, the criticism of the Referee motivated us to try to understand better what happens in homogeneous systems and whether we can find additional arguments for an existence of an intermediate scale at which the Navier-Stokes physics holds. To address these important points we have added a new section where we employ the nonlinear fluctuating hydrodynamics to show that due to the proximity to integrable model, there is a clear crossover between Navier-Stokes hydrodynamics (with transport coefficients found in this work) and KPZ superdiffusion, which indeed emerges at the largest scales in the system. While the crossover length scale, on which NS equations emerge from GHD scales like $ \sim \lambda^{-2}$, the crossover to KPZ happens at even larges length scales $\sim \lambda^{-4}$. Thus, there is a relevant window of length scales, in which NS hydrodynamics with finite transport coefficients is the correct hydrodynamic theory of the system. We summarize these findings in Fig.1 in the text. The importance of these results is reflected in a modified title and abstract of the article. They highlight the fact that our work addresses now the important issue of emergence of superdiffusion in the 1d fluids.

We do agree with the Referee that transport coefficients as given by Green-Kubo formula are divergent, this fact is predicted by nonlinear fluctuating hydrodynamics. However, it is important here to distinguish between Green-Kubo transport coefficients (which are given by integral from $t=0$ to $t=\infty$, thus probing the largest scales in the system) and bare transport coefficients, which enter the equations of nonlinear fluctuating hydrodynamics through diffusion matrix $D$. We stress that in order to arrive at long-time tails and divergent Green-Kubo integrals one has to start with finite bare transport coefficients. The Chapman-Enskog theory provides explicit formulas for the bare transport coefficients and analysis in Sec. 5 clearly shows that they are crucial at intermediate (which can be very large in practice) length scales, on which mode-coupling terms are dominated by the diffusion. As we discuss in the text, the existence of such a crossover is essentially due to the fact that nearly-integrable systems are characterized by parametrically long (in integrability breaking parameter $\lambda$) relaxation time, which leads to large diffusion constants. This is the only necessary requirement, which opens the window for stable NS regime in the system. Thus, our results are perhaps of more general importance than the specific setup of GHD-Boltzmann equation. In the text we also shortly refer to other works, which found similar crossover between normal and superdiffusive transport in nearly-integrable systems.

We believe that our new results correctly address the points made in the report and that the new version warrants publication in SciPost Physics, instead of SciPost Physics Core.

Sincerely,
Authors

---

## Round 2 · Referee Report · Anonymous (Referee 2) · 2025-4-29

Report

I commend the authors for their detailed and thoughtful engagement with my comments. I agree with the authors that their new results predicting an explicit crossover timescale from a “conventional” hydrodynamic regime to an anomalous regime are novel and potentially of broader interest than the results reported in the initial draft. In particular, it was not at all clear from the conclusion of the previous draft that a parametrically large crossover region between normal and anomalous hydrodynamics could exist. The authors have now provided a thorough and detailed argument that such a regime does in fact exist. This both a posteriori justifies the Chapman-Enskog approach taken by the authors, and strikes me as interesting, physically insightful and concrete enough to be tested in future work. I am therefore happy to recommend the revised manuscript for publication in Scipost.

I did have a couple of follow-up remarks that the authors might consider:

  1. Chapman-Enskog is indeed expected to yield accurate (if uncontrolled) results in weakly coupled 3D systems. But this a very different regime of approximation from 1D and strong coupling, which is why the authors’ predictions are still a hypothesis to be tested. This limitation of Chapman-Enskog is not really addressed in the text.

  2. The terminology of “GHD-Boltzmann” equations sounds redundant to me. The viewpoint that the GHD equation should be viewed as a Boltzmann equation without dissipative collision terms was explicitly proposed in Phys. Rev. B 97, 045407 (2018) and Phys. Rev. Lett. 120, 045301 (2018) and is implicit in Ref. 29. The authors also remark that “Although GHD-Boltzmann equation differs from the standard Boltzmann counterpart by the presence of nonlinear effective velocity in the streaming term and by diffusion of quasiparticles” but in fact both of these effects have been standard for decades in the kinetic theory of Fermi liquids, which is introduced e.g. in Landau and Lifshitz Vol. 9, Chapter 1.

Recommendation

Publish (surpasses expectations and criteria for this Journal; among top 10%)

---

## Round 2 · Referee Report · Anonymous (Referee 1) · 2025-5-3

Report

The authors made substantial additions to the paper, which address the concerns I raised before. I find their results regarding the crossover between normal and anomalous hydrodynamic behaviour especially interesting. Therefore, I have no hesitation in recommending their work for publication in Scipost Physics.

Recommendation

Publish (easily meets expectations and criteria for this Journal; among top 50%)

---

## Round 2 · Author Response

Dear Editor,

Thank you for taking care of our submission.

We are aware that our manuscript has received criticism from the two Referees. However, we are convinced that with this revised manuscript we were able to fully address the points raised in both reports. Therefore, we would like to ask you to reconsider submitting our article once again to the SciPost Physics.

Importantly, motivated by the comments of both Referees, we have decided to expand our article and include a new section about the emergence of anomalous transport in 1d fluids. This section contains essential new results, which are of major importance for the topic of weak integrability breaking and address the doubts of the Second Referee. Their objection assumes that the transport in 1d is always anomalous despite the counterexamples that we gave in the Conclusions to the first version of this manuscript. Those counterexamples were pointing towards a crossover regime between a normal and anomalous hydrodynamics. In this new section we present a general argument for the existence of the crossover by employing the non-linear fluctuating hydrodynamics. This is the very method that was used to argue for the anomalous hydrodynamics in the first place and shows that there is no contradictions in existence of these two regimes. The relevance of the additional content is reflected in the modified title of the manuscript, which highlights now also the crossover between diffusive and superdiffusive transport.

We strongly believe that our article, due to these improvements, fulfills the criteria of the journal. We hope that the Referees, once they see the recent changes, will agree with us on that point.

Sincerely,
Authors

---

## Round 2 · List of Changes

1. We changed the title of our manuscript to "Chapman-Enskog theory and crossover between diffusion and superdiffusion for nearly integrable quantum gases".
2. We added a new section (Sec.5 in the present version) which discusses the crossover between Navier-Stokes diffusive transport and anomalous, Kardar-Parisi-Zhang superdiffusion.
3. We modified the abstract which presents now a short summary of the new section.
4. We modified the introduction and the summary to incorporate the new findings of the second version.
5. We added Figure 1, which presents crossovers between different hydrodynamic regimes of nearly integrable quantum gases.
6. We added Figure 2 presenting numerical solutions to the mode-coupling equations.
7. We added Appendix B with material related to the content of new section.

List of corrections:

1. We introduced abbreviation for "Chapman-Enskog" as "ChE" and "Navier-Stokes" as "NS"
2. We corrected typos in Eqs. (15), (127).
3. We updated the references which in present enumeration are [30], [51] and added references [66], [80], [81], [82], [84], [88].

---

## Editorial Decision

published